# Detection and manipulation of live antigen-expressing cells using conditionally stable nanobodies

Jonathan CY Tang[1,2†], Eugene Drokhlyansky[1,2†], Behzad Etemad[3], Stephanie Rudolph[4], Binggege Guo[1,2], Sui Wang[1,2], Emily G Ellis[4], Jonathan Z Li[3], Constance L Cepko[1,2*]

[1]Department of Genetics, Howard Hughes Medical Institute, Harvard Medical School, Boston, United States; [2]Department of Ophthalmology, Howard Hughes Medical Institute, Harvard Medical School, Boston, United States; [3]Brigham and Women's Hospital, Harvard Medical School, Boston, United States; [4]Department of Neurobiology, Harvard Medical School, Boston, United States

**Abstract** The ability to detect and/or manipulate specific cell populations based upon the presence of intracellular protein epitopes would enable many types of studies and applications. Protein binders such as nanobodies (Nbs) can target untagged proteins (antigens) in the intracellular environment. However, genetically expressed protein binders are stable regardless of antigen expression, complicating their use for applications that require cell-specificity. Here, we created a conditional system in which the stability of an Nb depends upon an antigen of interest. We identified Nb framework mutations that can be used to rapidly create destabilized Nbs. Fusion of destabilized Nbs to various proteins enabled applications in living cells, such as optogenetic control of neural activity in specific cell types in the mouse brain, and detection of HIV-infected human cells by flow cytometry. These approaches are generalizable to other protein binders, and enable the rapid generation of single-polypeptide sensors and effectors active in cells expressing specific intracellular epitopes.

*For correspondence: cepko@ genetics.med.harvard.edu

†These authors contributed equally to this work

## Introduction

Many applications in biology and medicine require the ability to target a subset of cells in a population based upon specific cellular characteristics. Although this can be achieved by exploiting transcriptional elements that are selectively active in a subset of cells, specific elements are often not available, and can be difficult to generate. Alternatively, other features that distinguish cells, such as expression of a specific RNA or protein, may be exploited. Recently, it has become possible to utilize specific intracellular proteins to drive desired molecular events, using RNA-based binders in cells (*Auslander et al., 2014*; *Culler et al., 2010*; *Kennedy et al., 2014*; *Saito et al., 2011*), and protein-based binders in cells and animals (*Tang et al., 2013*, *2015*). While current methods are promising, protein-responsive systems are continually evolving. There remains a need for generalizable strategies that enable rapid conversion of diverse classes of binders into protein-responsive tools.

Antibodies are widely adopted reagents used for protein detection and manipulation. Their popularity derives from their superior specificity and high affinity, achieved in large part by the stringent selection in an immunized animal. Nbs, the antigen recognition portions of single chain antibodies found in camelids (*Hamers-Casterman et al., 1993*) and cartilaginous fishes (*Greenberg et al., 1995*), bind their cognate antigens with high affinity and specificity, and have the added advantage over heterotetrameric antibodies in that they are very stable in the intracellular environment. Fusions

**eLife digest** Biologists often wish to study the role of a particular cell type within an organism, but such studies are often not possible due to the lack of reagents that allow one to gain control of the cell type of interest. One method that can be used to detect and manipulate the cells that express specific proteins uses molecules called antibodies. An antibody can strongly bind to a specific part of a protein, and a diversity of antibodies that bind to different proteins can be isolated by animal immunization, or by using molecular or cell-based methods.

Antibodies from camelid species (which include camels and llamas) are increasingly being used to detect and manipulate proteins in living cells. The variable region of these antibodies – also known as the nanobody – recognises the proteins that the antibody binds to, and often just this fragment of the antibody is used in protein detection experiments. However, nanobodies are stable even in cells that do not contain their target proteins, which makes it difficult to use nanobodies to study just a specific cell type within an organism.

Tang, Drokhlyansky et al. have now developed a way of engineering the sequence of a nanobody so that it is broken down in living cells unless it is bound to its protein target inside the cell. Any protein that is tethered to the engineered nanobody is also broken down. For example, some tethered proteins with useful biological activities are fluorescent proteins and enzymes that can modify DNA. When one of these engineered nanobodies binds to a protein target of interest, the activity of the nanobody-tethered protein can be turned on in just those cells that produce the targeted protein. Thus, this strategy of engineering allows "conditionally stable" tools to be generated.

A core set of sequence alterations can be used to modify different nanobodies that target different proteins. Tang, Drokhlyansky et al. have demonstrated the uses of several of the resulting conditionally stable nanobodies. In one application, the nanobodies were used to target specific cell types in the mouse brain in a way that allowed the activity of these cells to be controlled by light. Another application of the technique enables live human cells that have been infected with HIV to be detected and isolated.

The conditionally stable nanobody tools can be used to detect and manipulate cells that express any protein for which a camelid antibody exists. Tang, Drokhlyansky et al. therefore hope that biologists who work in a wide range of fields will find the tools useful for studying many different types of organisms and biological processes.

between Nbs and proteins with desirable activities have enabled a number of applications in living cells (*Caussinus et al., 2012*; *Irannejad et al., 2013*; *Kirchhofer et al., 2010*; *Rothbauer et al., 2006*; *Tang et al., 2013*, *2015*). Despite these successes, it has been difficult to take advantage of Nbs for live cell applications requiring cell-specificity, as genetically expressed Nb-fusion proteins are stable and active even in cells that do not express the cognate antigens. This is a general problem that applies to any class of protein-based binder. To address this issue, we reasoned that the Nb portion of a single Nb-fusion protein could be modified to be conditionally stable in living cells, with stability conferred by the presence of antigen (*Figure 1A*). A similar approach has been used to create small molecule-dependent domains, for temporal control or tuning of protein activity (*Banaszynski et al., 2006*).

Here, we report the isolation of destabilized Nbs (dNbs) using a strategy that should be generalizable to other types of protein-based binders. We isolated a dNb whose destabilizing mutations fell within the structurally conserved framework region of Nbs. These destabilizing mutations could simply be transferred to other Nbs to rapidly generate antigen-dependent stability. dNbs were able to destabilize fusion partners having a variety of activities, including fluorescent proteins, site-specific recombinases and genome editing enzymes. We used these reagents to optogenetically control neural activities in specific cell types, as well as detect and isolate Human immunodeficiency virus (HIV) infected cells based upon the expression of the HIV-1 capsid protein. Thus, this work offers a generalizable strategy to label and manipulate specific cell populations in cellular and animal systems, with specificity endowed by protein expression and/or specific cellular features.

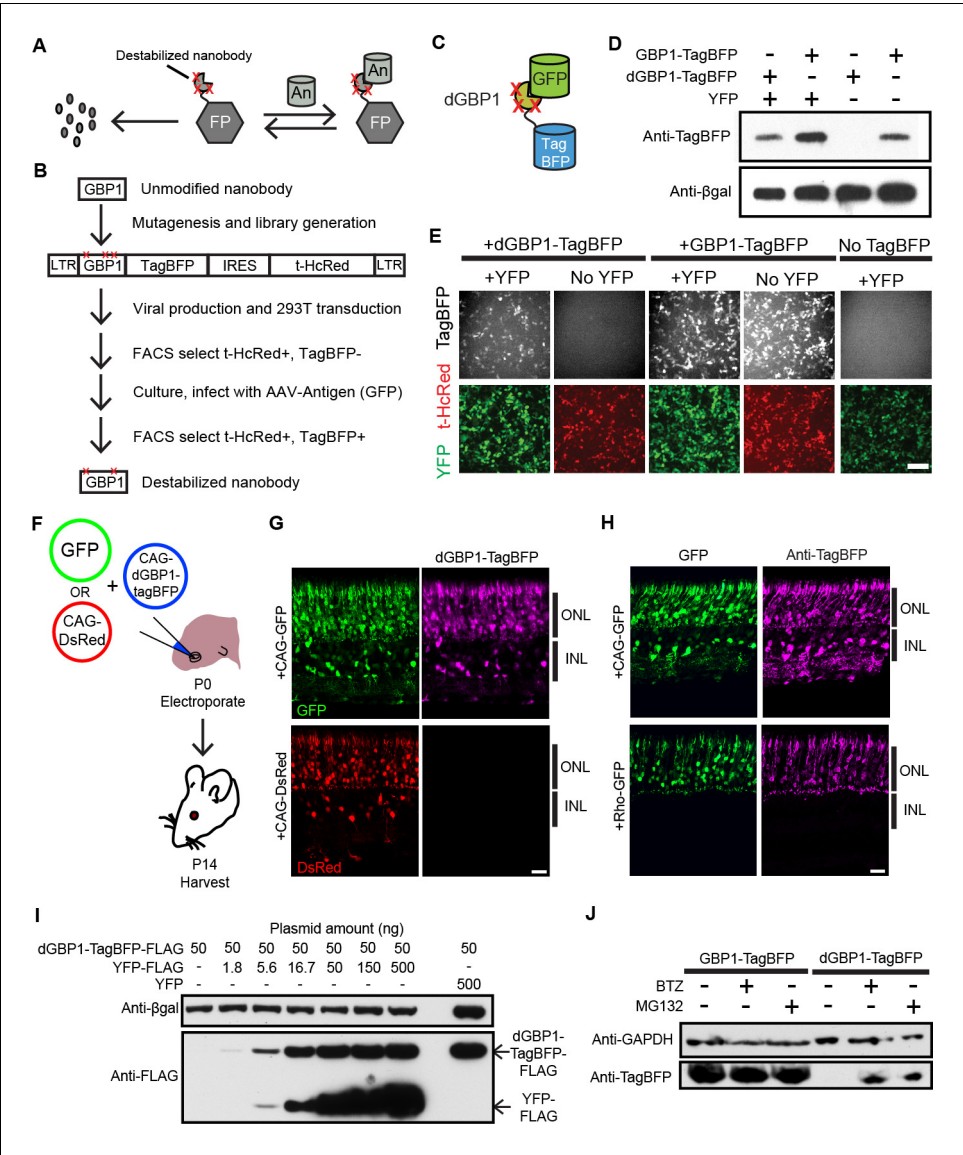

**Figure 1.** Isolation of a destabilized Nb whose protein level depends upon antigen co-expression. (**A**) Concept of antigen-controlled protein stabilization. FP, fusion protein; An, antigen. (**B**) Strategy for isolating dNbs. LTR, long terminal repeat. (**C**) GFP-dependent stabilization of dGBP1 tagged with TagBFP. (**D**) Western blot of transfected 293T cell lysate for TagBFP and βgal (a transfection control) shows that the level of dGBP1-TagBFP is dependent upon YFP co-expression. YFP is a derivative of GFP. (**E**) YFP (green)-dependent dGBP1-TagBFP fluorescence in transfected 293T cells. t-HcRed (red) is a transfection marker for cells with TagBFP (gray) fusion constructs. Scale bar, 100 μm. All results from (**D, E**) are representative of three independent experiments, for 3 biological replicates. (**F**) Schematic of electroporation experiment. (**G**) GFP (green), but not DsRed (red), promotes dGBP1-TagBFP fluorescence (magenta) in the mouse retina. ONL, outer nuclear layer; INL, inner nuclear layer. (**H**) The expression pattern of TagBFP protein, as detected by Anti-TagBFP (magenta), can be altered by changing the GFP expression pattern with broadly active (CAG) or rod photoreceptor-specific (Rho) promoters. TagBFP immunodetection was not carried out in (**B**) as the TagBFP antibody cross-reacts with DsRed. Scale bar is 20 μm. Biological replicates (retinas): n = 3 for all conditions. (**I**) Dependence of dGBP1-TagBFP protein level on YFP dose in transfected 293T cells. (**J**) The ubiquitin proteasome pathway is involved in degradation of dGBP1. Transfected 293T cells were treated with drugs for 20 hr. BTZ, Bortezomib. All results from (**I, J**) are representative of three independent experiments, for 3 biological replicates. Additional data related to characterizations of dGBP1-TagBFP in vitro and in vivo are shown in *Figure 1—figure supplements 1–2*.

The following figure supplements are available for figure 1:

*Figure 1 continued on next page*

*Figure 1 continued*

**Figure supplement 1.** YFP-dependent dGBP1-TagBFP stabilization in cells.

**Figure supplement 2.** Detection of antigen-expressing cells with dNb in vivo.

## Results

### Isolation and characterization of a destabilized Nb

To test whether it is possible to modify an Nb such that its intracellular protein level is strongly dependent upon antigen co-expression, we used the GFP-binding Nb, GBP1, for proof-of-concept experiments (*Kirchhofer et al., 2010*; *Rothbauer et al., 2006*) (*Figure 1B,C*). We generated a Moloney murine leukemia virus (MMLV) library encoding randomly mutagenized variants of GBP1 fused to the blue fluorescent protein, TagBFP (*Subach et al., 2008*). t-HcRed (*Gurskaya et al., 2001*) was co-expressed via an IRES to report infection. TagBFP and t-HcRed bear little amino acid similarity to Aequorea-derived GFP and its derivatives. We infected 293T cells with this library, and combined FACS with super-infection by a GFP-encoding recombinant adeno-associated virus (rAAV) to isolate GBP1-TagBFP variants whose blue fluorescence depended upon GFP expression (*Figure 1B*; Materials and methods). One hundred GBP1 variants were then individually screened for enhanced TagBFP expression in the presence of yellow fluorescent protein (YFP), a GFP derivative known to also interact with GBP1 (*Rothbauer et al., 2008*; *Tang et al., 2013*). Some variants showed fusion TagBFP aggregates within well-transfected cells when YFP was absent, but became soluble in the cytoplasm when YFP was present (*Figure 1—figure supplement 1A*). Notably, a GBP1 variant carrying 6 amino acid changes (A25V, E63V, S73R, S98Y, Q109H, S117F) gave little to no TagBFP fluorescence, with no signs of aggregation in the absence of YFP. We focused our efforts on this variant, which will hereafter be referred to as destabilized GBP1 (dGBP1). dGBP1-TagBFP showed strong fluorescence and protein level when co-expressed with GFP or YFP, but became weakly detectable or undetectable when antigen was absent (*Figure 1D,E* and *Figure 1—figure supplement 1*). In contrast, unmodified GBP1-TagBFP showed strong fluorescence and protein level regardless of antigen co-expression (*Figure 1D,E*). Interestingly, we detected an increase in the level of wildtype GBP1-TagBFP protein in the presence of YFP (*Figure 1—figure supplement 1C*). In an electroporation experiment using the mouse retina, dGBP1-TagBFP fluorescence and protein level were detected only upon GFP co-expression in vivo (*Figure 1F–H*, *Figure 1—figure supplement 2*). Strikingly, the efficiency of TagBFP stabilization by GFP expression was nearly 100%, i.e. almost every GFP+ cell was TagBFP+ (*Figure 1—figure supplement 2*). The efficiency of the TDDOG and CRE-DOG systems was, at the highest, ~60% in similarly designed electroporation experiments (*Tang et al., 2013*, *2015*). This difference likely reflects the requirement for delivery of a greater number of components for T-DDOG and CRE-DOG experiments. Taken together, these data show that one can create a highly destabilized Nb whose protein level is dependent upon co-expression with its cognate antigen in vitro and in vivo.

We previously created Nb-based, antigen-dependent systems that use the antigen as a scaffold for the assembly of split protein domains or fragments (*Tang et al., 2013*, *2015*). Complex assembly can be inhibited when excessive antigen levels saturate antigen-binding sites in Nb-fusion proteins (*Tang et al., 2013*, *2015*). In contrast, a single polypeptide, dNb-fusion protein should not suffer the same limitation. Indeed, YFP promoted dGBP1-TagBFP stability in a dose-dependent manner, with no adverse effects even when YFP plasmid was transfected at ten-fold excess of dGBP1-TagBFP plasmid (*Figure 1I*). To investigate the mechanism of dNb destabilization, dGBP1-TagBFP-transfected 293T cells were treated with the ubiquitin proteasome inhibitors, MG132 or Bortezomib (BTZ) (*Kisselev et al., 2012*). dGBP1-TagBFP protein was evident following addition of either inhibitor and was absent without inhibitors, indicating that it was degraded by the ubiquitin proteasome system (UPS) (*Figure 1J*).

## Generation of additional dNbs by mutation transfer

Discosoma-derived mCherry fused to dGBP1 (dGBP1-mCherry) also showed antigen-dependent sta-bilization. Unlike dGBP1-TagBFP (*Figure 1E*), some aggregation of dGBP1-mCherry occurred inside cells when antigen was absent (*Figure 2—figure supplement 1*). We use dGBP1-mCherry as a sensi-tized reporter to map the key residues involved in GBP1 stability, by comparing the level of fluores-cence and aggregation of the fusion proteins in cells. (*Figure 2—figure supplements 1,2*). C/S98Y and S117F showed strong destabilizing effects, as seen in both sufficiency and necessity experi-ments. S73R and Q109H also had destabilizing effects in single mutant analyses. GFP rescued the destabilization phenotype of all mutants. Thus, specific single dGBP1 mutations had clear destabiliz-ing effects, which could be enhanced by combination with other destabilizing mutations.

The dGBP1 mutations mapped onto the structurally conserved framework regions of Nbs (*Muyl-dermans, 2013*), and 99–100% of Nbs (n=76) shared the same residue as GBP1 at each of the 3 most destabilizing positions (3maj: S73R, C/S98Y, S117F) (*Figure 2A*; *Figure 2—figure supplement 3A*). Further, a survey across 76 unique Nb-antigen interfaces, gathered from a total of 102 crystal structures, indicated that Nb positions corresponding to those of dGBP1 A25V, S73R, S98Y and S117F were universally located outside of all Nb-antigen interfaces (*Figure 2—figure supplement 3B*). Nb positions corresponding to dGBP1 Q109H fell outside of 99%, or 75 of 76 unique Nb-anti-gen interfaces. Positions equivalent to dGBP1 E63V were found in 22%, or 17 of 76 unique Nb-anti-gen interfaces, and in close proximity to the interface in 9%, or 7 of 76 of the cases. Given these results, we hypothesized that the destabilizing framework mutations could be transferred across Nbs to rapidly create antigen-dependent stability. We transferred all dGBP1 mutations (6mut) and the 3maj mutations to Nbs targeting the HIV-1 capsid protein (αCA) and *Escherichia coli* dihydrofolate reductase (αDHFR), respectively. dNbs created by mutation transfer behaved similarly as dGBP1 in that TagBFP fusion fluorescence and protein level both depended upon expression of the cognate antigen (*Figure 2B–E*). Destabilization also depended on degradation by the UPS (*Figure 2F*). We then explored whether dNb-TagBFP expression had an adverse effect on antigen expression. Using western blots to quantify protein levels, we found that dGBP1-TagBFP did not have an obvious effect on YFP protein level, when compared to the negative control condition whereby αCA$^{6mut}$-TagBFP replaced dGBP1-TagBFP (*Figure 2—figure supplement 3C*).

To further investigate the generality of the mutation transfer approach, we transferred the 3maj mutations to 9 Nbs that recognize epitopes of intracellular origin (*Figure 2G*; Materials and meth-ods). All dNb-TagBFPs showed strongly reduced fluorescence relative to their unmodified Nb coun-terparts and occasionally formed faint fluorescent punctae in cells over-expressing the fusion proteins (*Figure 2B,D and 2G*). Importantly, whereas no unmodified Nb showed >2 fold increase in TagBFP fluorescence in response to antigen co-expression, 8 of 9, or 89% of dNbs, passed this threshold (*Figure 2G*). Notably, mutations that destabilized a *Camelus dromedarius* Nb (GBP1) had very similar effects on Nbs derived from *Vicugna pacos* and *Llama glama*, indicating that destabiliz-ing mutations can be transferred across Nbs from different camelid species to create antigen-depen-dence (*Figure 2A and G*; *Table 1*).

## Generation, optimization and uses of antigen-specific effectors

To explore if additional fusion proteins could be rendered antigen-dependent, we engineered dNbs onto two popular site-specific recombinases, Cre and codon-optimized Flp (Flpo) (*Luo et al., 2008*; *Raymond and Soriano, 2007*). Both enzymes were rendered GFP-dependent after fusion to dGBP1, but not GBP1 (*Figure 3A–B*). By increasing the number of dGBP1 domains fused to either enzyme, residual GFP-independent recombination of the initial fusions was decreased without affecting GFP-dependent recombination (*Figure 3A–B* and *Figure 3—figure supplement 1A–B*). Notably, Flpo fused to tandemly repeated dGBP1 (dGBP1x2-Flpo, or Flp dependent on GFP [Flp-DOG]) had insig-nificant background signal and over 600 fold induction by GFP (*Figure 3B*). We also rapidly con-structed an Flpo fusion protein dependent upon the C-terminal portion of HIV-1 CA (C-CA) using mutation transfer. This construct was functional in vitro and in vivo(*Figure 3C* and *Figure 3—figure supplement 1D*). Further, both GFP- and C-CA-dependent Flpo responded to antigen in a dose-dependent manner (*Figure 3—figure supplement 1C–D*). Thus, dNbs can confer antigen-depen-dent control over different types of effector proteins, and can adequately reduce the activity of a highly sensitive enzyme when antigen is absent.

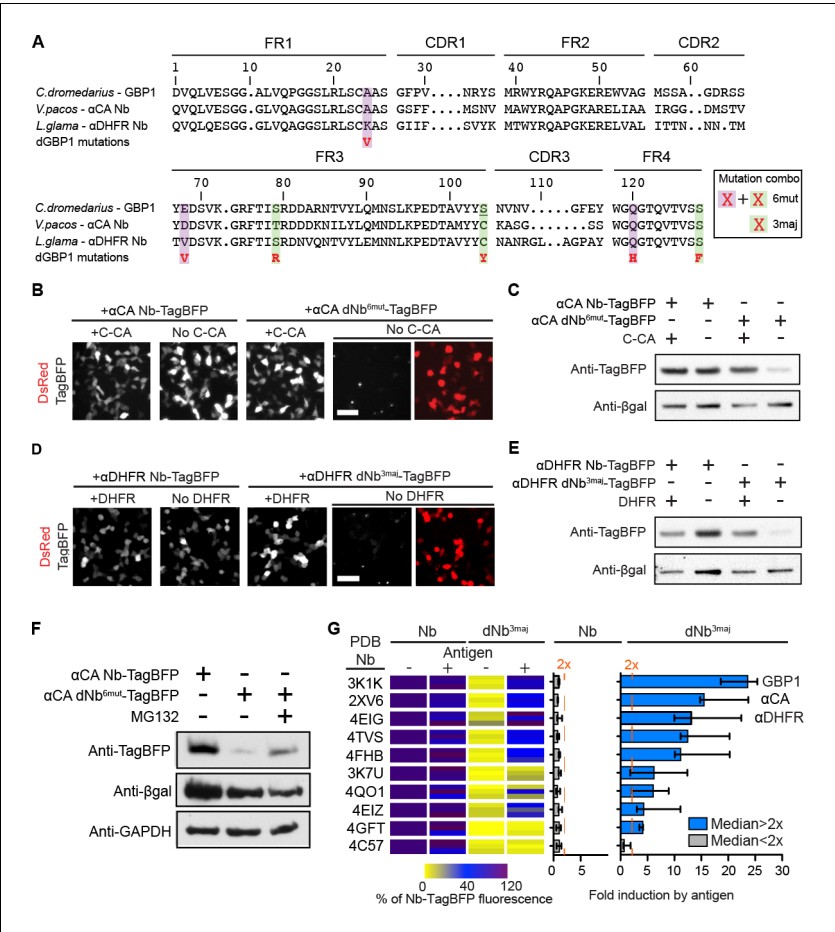

**Figure 2.** dGBP1 destabilizing mutations can be transferred to Nbs derived from different species to create antigen-dependent stability. (**A**) Protein alignment of Nbs against GFP (GBP1), HIV-1 CA (αCA) and *E.coli* DHFR (αDHFR). Amino acid positions numbered according to the ImMunoGeneTics information system (IMGT). FR, framework; CDR, complementarity determining region. Purple- and green-highlighted residues indicate dGBP1 mutation position. Green-highlighted residues (3maj) are most destabilizing when mutated. The underlined serine was a cysteine in the original GBP1. (**B–E**). Transfer of dGBP1 mutations to other Nbs. Destabilized, but not unmodified αCA and αDHFR showed antigen-dependent fluorescence (**B,D**) as well as protein level (**C,E**). DsRed (red) indicates transfected cells in (**B,D**) and is only shown when TagBFP (gray) results are negative. Scale bar, 50 μm. (**F**) A dNb generated by mutation transfer was degraded by the UPS. αCA-dNb^6mut-TagBFP showed an increase in protein level when transfected 293T cells were treated with MG132 for 6 hr. Results representative of 3 independent experiments. (**G**) Heat map showing median TagBFP fluorescence intensity of Nb-TagBFP fusions. All Nbs shown recognize epitopes of intracellular origin. Fluorescent readings were normalized to that of unmodified Nb (no antigen) condition, which was set to 100. n = 3 biological replicates pooled from 3 independent experiments per Nb. Each biological replicate result is shown as a horizontal bar in the heat map. Bar graphs indicate median and maximum-to-minimum range. Additional data related to mutation transfer of destabilizing mutations are shown in *Figure 2—figure supplements 1–3*. Source data for % Nb-TagBFP fluorescence values are shown in *Figure 2—source data 1*.

The following source data and figure supplements are available for figure 2:

**Source data 1.** Source data for fluorescence quantifications of Nb-TagBFP tests.
**Figure supplement 1.** Mapping of mutations necessary for dGBP1 destabilization.
**Figure supplement 2.** Mapping mutations sufficient for dGBP1 destabilization.
**Figure supplement 3.** Generation of dNbs by mutation transfer.

**Table 1.** List of tested nanobodies and their associated antigens.

| Nb PDB code | Species of origin | Tested antigen | Antigen species/ pathogen | Endogenous location of epitope |
|---|---|---|---|---|
| 3K1K | *C.dro* | GFP | *Aequorea victoria* | Intracellular |
| | | YFP | *Aequorea victoria* | Intracellular |
| | | YFP-FLAG | *Aequorea victoria* | Intracellular |
| 2XV6 | *V.pacos* | Capsid protein p24 C-terminal domain, residues 278-352 of gag polyprotein | HIV-1 | Intracellular |
| | | Capsid protein p24 | HIV-1 | Intracellular |
| 4EIG | *L.glama* | Dihydrofolate reductase | *Escherichia coli* | Intracellular |
| 4TVS | *V.pacos* | Torsin-1A-interacting protein 1, UNP residues 356-583 | *Homo sapiens* | Intracellular |
| 4EIZ | *L.glama* | Dihydrofolate reductase | *Escherichia coli* | Intracellular |
| 4FHB | *L.glama* | Dihydrofolate reductase | *Escherichia coli* | Intracellular |
| 4QO1 | *L.glama* | Cellular tumor antigen p53 DBD, UNP residues 92-312 | *Homo sapiens* | Intracellular |
| 3K7U | *L.glama* | MP18 RNA editing protein | *Trypanosoma brucei* | Intracellular |
| 4GFT | *L.glama* | Myosin A tail domain interacting protein C-terminal domain, UNP residues 137-204 | *Plasmodium falciparum* | Intracellular |
| 4C57 | *L.glama* | Cyclin-G associated kinase, kinase domain residues 14-351 | *Homo sapiens* | Intracellular |

C.dro: Camelus dromedarius; L.glama: Lama glama; V.pacos: Vicugna pacos.

The ability to control Flpo activity with tandem dNb-fusions raised the possibility of imposing dual regulation on effector protein activity using two dNbs, each targeting a different antigen (*Figure 3D–E* and *Figure 3—figure supplement 1E*). We tested this by generating Flpo fused to dGBP1 and αCA dNb$^{6mut}$ (dGC-Flpo) or dGBP1 and αDHFR dNb$^{3maj}$ (dGD-Flpo). Strong Flpo recombination was triggered only when both antigens were present (*Figure 3D–E*). This shows the feasibility of using dNb-fusion proteins to create synthetic circuits whereby dual inputs are integrated entirely at the protein level.

We further tested whether it was possible to perform genome targeting and editing under the control of specific antigen(s) (*Figure 4A*). We created a fusion between two αCA dNb$^{6mut}$ and Cas9 (dCC-Cas9) and delivered the construct to an engineered human cell line that expresses β-galactosidase upon removal of a loxP-stop-loxP cassette. We also delivered a guide RNA that can specifically target the loxP sites, leading to Cas9-mediated deletion of the stop cassette (dCC-Cas9-loxPgRNA) (*Figure 4B–C*). Co-expression of C-CA with dCC-Cas9 and loxPgRNA triggered genome-editing events, while little to no β-galactosidase expression was detected when C-CA was absent (*Figure 4D*). The efficiency of C-CA-dependent genome editing approached that of control Cas9 (*Figure 4D*). This result demonstrated the feasibility of using intracellular epitopes to initiate genome editing or targeting.

## Retrofitting transgenic GFP mouse lines for cell-specific manipulation of gene expression and neural activity

To evaluate the usefulness of dNbs, we applied dNb-fusion proteins in situations where we could not control antigen level. GFP and its derivatives (*Tsien, 1998*) are widely used to label cell types, with specificity dependent upon cellular features such as gene transcription (*Chalfie et al., 1994*) or neuronal connectivity (*Beier et al., 2011*; *DeFalco et al., 2001*; *Ekstrand et al., 2014*; *Lo and Anderson, 2011*; *Wickersham et al., 2007*). Genetic manipulation of GFP-labeled cells can reveal their functions, but current approaches require delivery of 2 or more αGFP Nb-fusion proteins (*Tang et al., 2015*, *2013*). Here, we used electroporation and rAAV to deliver the one-component Flp-DOG along with Flp-dependent constructs to the retinas of Tg(CRX-GFP) (*Samson et al., 2009*) and cerebella of Tg(GAD67-GFP) (*Tamamaki et al., 2003*) lines, respectively. In both instances, robust Flpo recombination was detected in GFP+ tissues, but not in GFP-negative tissues labeled with electroporation, infection or injection markers (*Figure 5A–C*; *Figure 5—figure supplement 1*,

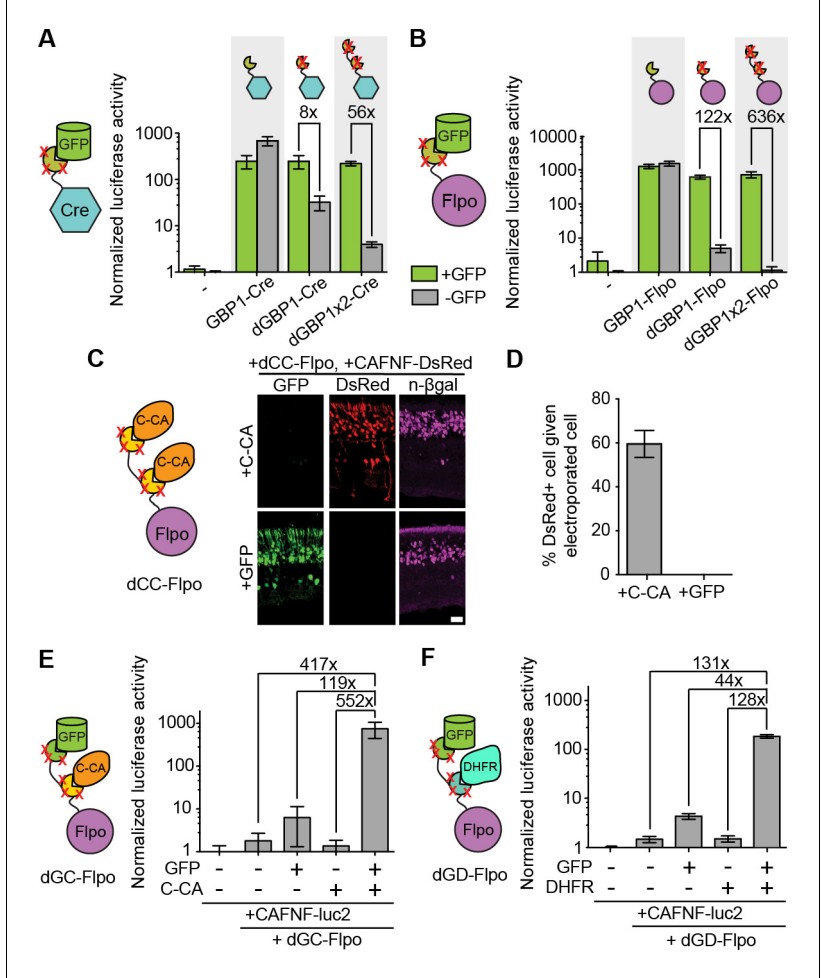

**Figure 3.** Generation and optimization of antigen-specific effectors based on dNbs. (A) Reporter assay of transfected 293T cells testing dGBP1-Cre fusion constructs for activation of a Cre-dependent luciferase construct. (B) Reporter assay of transfected 293T cells testing dGBP1-Flpo fusion constructs for activation of a Flpo-dependent luciferase construct. Cells were harvested at 15 hr (A) or 36 hr (B) post-transfection. All results are representative of 3 independent experiments. Sample size per condition, n = 12 (A) and n = 18 (B). (C) Rapid generation of a C-CA-dependent Flpo by transfer of dGBP1 mutations to αCA Nb. Schematic of C-CA-dependent Flpo (left). C-CA can promote Flpo recombination in the mouse retina (right). n-βgal was an electroporation marker. Scale bar, 20 μm. (D) Quantification of C-CA-dependent Flpo activity. In (C,D), 4 and 3 electroporated retinas were analyzed for +C-CA and +GFP conditions, respectively. (E,F) Reporter assay in 293T cells using Flp-dependent luc2 construct, CAFNF-luc2. Results show dependence of Flpo activity on the presence of both GFP and C-CA (E) or GFP and DHFR (F). n = 6 per condition in (E) and n = 9 per condition in (F). Consistent results were obtained in 3 independent experiments. All plots indicate mean and standard deviation. Additional characterization of dNb-based effectors is shown in *Figure 3—figure supplement 1*.

The following figure supplement is available for figure 3:

**Figure supplement 1.** Additional information on antigen-specific effectors.

and *Figure 5—figure supplement 2A–E*). We used rAAV-delivered Flp-DOG to induce ChR2-mCherry expression in GABAergic Purkinje cells (PCs) of Tg(GAD67-GFP) cerebella (*Figure 5C*). Under conditions in which infection did not alter spontaneous firing frequency or input resistance, we evoked excitatory photocurrents and synaptic inhibitory currents in ChR2-mCherry+ PCs, with inhibitory inputs from neighboring ChR2-mCherry+ neurons that contacted the recorded PCs (*Figure 5*; *Figure 5—figure supplement 2F–G*). GFP+ neurons that did not express ChR2-mCherry, as

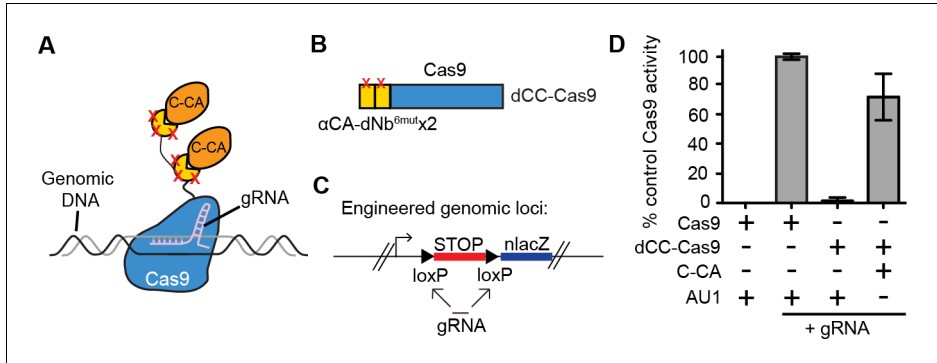

**Figure 4.** Intracellular antigen can trigger genome editing via dNbs. (**A**) Schematic of Cas9 fusion protein inducible by C-CA binding. (**B**) Fusion configuration of tandemly repeated C-CA Nb to Cas9, giving dCC-Cas9. (**C**) dCC-Cas9 activity was assayed for βgal expression in a human TE671 cell line engineered to contain a lacZ reporter inactive in expression due to a loxP-STOP-loxP transcriptional termination cassette. gRNA targets both loxP sequences for Cas9 cleavage and STOP removal. (**D**) dCC-Cas9 shows C-CA-dependent activity. Cas9 activity is represented as number of βgal+ cells induced as a percentage of unfused Cas9 activity (100%). AU1 is used as a negative control peptide, expressed in place of C-CA. Plots were mean ± standard deviation. n = 3 or 5 biological replicates (transfected wells) per condition. Consistent results were obtained in 3 independent experiments.

---

well as control ZsGreen+ neurons in GFP-negative animals, never showed light-evoked photocurrents, indicating antigen-specificity of the system (*Figure 5D*). Thus, Flp-DOG provides a much simpler approach to manipulate GFP-defined cell types over pre-existing methods. Overall, these results demonstrate that dNb-fusion proteins enable functional manipulation of antigen-expressing cells in vivo.

## Detection of HIV-1 reactivated human cells by flow cytometry

Next, we tested whether dNbs could be used to detect and isolate live, antigen-expressing cells. ACH-2 (*Folks et al., 1989*), a human T-cell line chronically infected with HIV-1, is widely used to study HIV-1 persistence (*Clouse et al., 1989*). Destabilized αCA fused to either TagBFP or TagRFP (*Matz et al., 1999*) were expressed in ACH-2 or the uninfected parental cell line (CEM), under conditions in which HIV-1 was reactivated with phorbol 12-myristate 13-acetate (PMA) (*Poli et al., 1990*) (*Figure 6A* and *Figure 6—figure supplement 1*). Using flow cytometry, we detected fluorescence of the destabilized fusions selectively in ACH-2, but not CEM cells (*Figure 6B–C* and *Figure 6—figure supplement 1B*). ACH-2-specific fluorescence was dependent upon CA recognition, as the effect was not observed with dGBP1-TagBFP. Importantly, unmodified αCA Nb-TagBFP fluoresced strongly in both cell lines and could not be used to distinguish between the two (*Figure 6B–C*). As positive controls, we confirmed that αCA dNb[6mut]-TagBFP could be stabilized by C-CA co-expression in CEM cells (data not shown) and that the HIV-1 CA antigen was specifically detected by immunofluorescence in ACH-2, but not CEM cells (*Figure 6—figure supplement 1C*). Thus, dNbs make possible detection of intracellular viral epitopes without the need for cell fixation or membrane permeabilization, enabling live monitoring of intracellular viral protein expression and specific isolation of infected live cells with a choice of spectrally distinct fluorescent proteins.

## Discussion

Here, we developed a straightforward and generalizable approach to convert a class of protein-based binders, the Nbs, into conditional reagents that can regulate biological activities within living cells that express a specific protein. Unlike conventional approaches, this cell-targeting strategy does not require knowledge of a cell-specific promoter, modifications of genomic loci, or modification of a cell-specific protein target. Instead, cell type-specificity is achieved by exploiting the intracellular protein expression pattern in cellular systems such as animals or pathogen-infected cells. These conditional reagents were isolated either by using the generalizable screening strategy

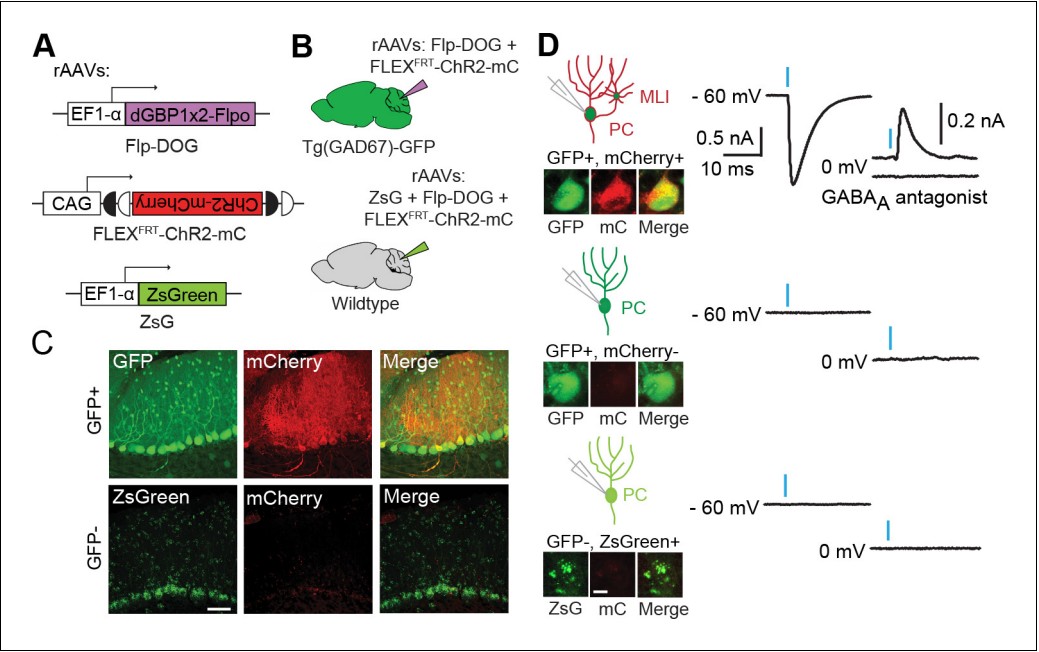

**Figure 5.** Applying Flp-DOG for optogenetic manipulation of transgenic GFP-labeled cell types in the mouse cerebellum. (**A**) rAAV reagents for (**B–D**). (**B**) Schematic showing delivery of rAAVs to the mouse cerebellum for cell type-specific manipulation in Tg(GAD67-GFP) animals. (**C**) Representative image showing that rAAV-encoded, GFP-dependent Flpo activates ChR2-mCherry expression selectively in the cerebellar cortex of Tg(GAD67-GFP) (n = 4), but not wildtype (n = 2) animals. ZsGreen is unrelated to GFP and was used as an infection marker for GFP-negative animals. Scale bar, 50 μm. (**D**) Optogenetic manipulation of GFP+, ChR2-mCherry+ cells. A pulse of blue light (blue bar) evoked a photocurrent at -60 mV holding potential, and an inhibitory synaptic current at 0 mV that was blocked by 5 μM of the GABA$_A$ receptor antagonist SR 95531, indicating activation of mCherry-ChR2+ cells synapsing onto the recorded PC. No photocurrents or synaptic currents were detected in GFP+/mCherry- PCs (n = 12 cells) from Tg(GAD67-GFP) animals as well as ZsGreen+/mCherry- PCs (n = 9 cells) from wildtype animals. Two-photon images show cells identified live for optogenetic manipulation and physiology. Bright ZsGreen aggregates were sometimes detected as faint signals in the mCherry channel. Scale bar, 10 μm. ZsG, ZsGreen; mC, mCherry. Additional data related to evaluation of Flp-DOG specificity for GFP expression are shown in *Figure 5—figure supplements 1* and *2*.

The following figure supplements are available for figure 5:

**Figure supplement 1.** Retrofitting transgenic GFP line for cell-specific gene manipulation in the mouse retina.

**Figure supplement 2.** Characterization of mouse cerebella infected with AAV-Flp-DOG.

described here, or by introducing a common set of destabilizing mutations to an ever-expanding repertoire of Nbs (*De Meyer et al., 2014*; *Fridy et al., 2014*). Previously, laborious screening was required to isolate pairs of Nb-fusion proteins that could reconstitute an activity when they co-occupied an antigen (*Tang et al., 2015*, *2013*). The ability to create single polypeptide, protein-responsive sensors and effectors greatly simplifies design, promotes generalizability, improves performance and enables easier delivery. These approaches should be generalizable to other classes of intracellular protein binders as well.

## Utility of antigen-specific sensors and effectors in cellular and animal systems

A key aspect of this study is the demonstration that these Nb-based, conditionally stable reagents were effective even when the experimenter did not choose the antigen levels. This was true for experiments conducted in human cell culture and in mice. Importantly, the detection of HIV-1 CA+ cells was achieved using a dNb that was rapidly generated by mutation transfer rather than by

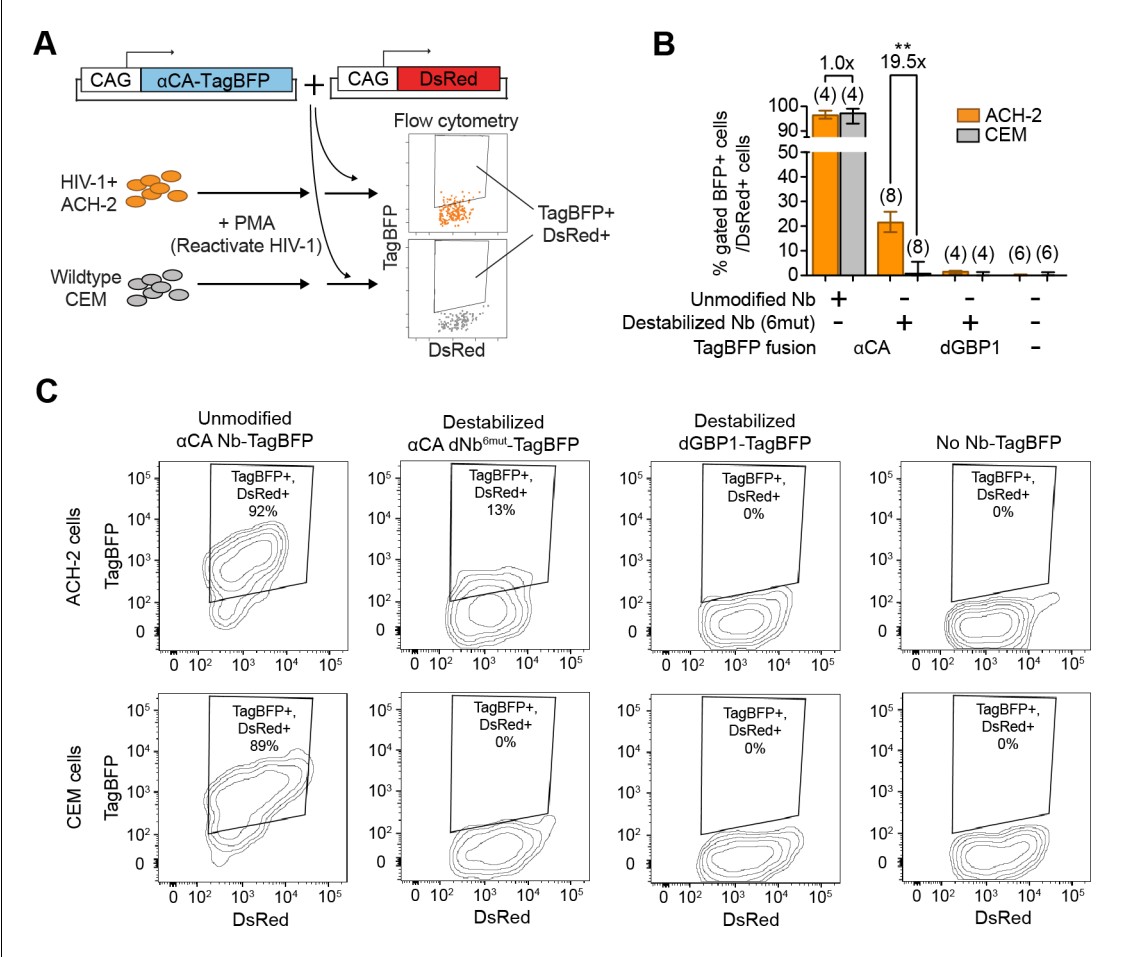

**Figure 6.** Detection of HIV-1 reactivated cells with a CA-specific sensor. (**A**) Schematic showing isolation of HIV-1 cells via flow cytometry using αCA-specific, dNb sensor. Both ACH-2 (HIV+) and CEM (HIV-) cells were treated with 10 nM PMA prior to transfection of sensors. CAG-DsRed was a transfection marker. (**B**) Destabilized, but not unmodified αCA Nb enabled selective isolation of reactivated HIV-1 cells using flow cytometry ($P = 0.0009$ for comparison between destabilized αCA, ACH-2 vs. CEM). Plot shows median and maximum-to-minimum range. The number of biological replicates (equal to number of independent experiments) for each condition is shown in parentheses. **p<$10^{-3}$, Mann-Whitney test. (**C**) Example of flow cytometry gating to isolate HIV-1 cells based on expression of CA. Cell populations are represented as log contour maps. Percentages of DsRed+ cells that were TagBFP+ are indicated for each condition. All cell populations were gated for DsRed expression. Results shown are representative of the number of biological replicates indicated in (**B**). Additional data related to HIV-1 sensors are shown in **Figure 6—figure supplement 1**.

The following figure supplement is available for figure 6:

**Figure supplement 1.** dNb sensor against HIV-1 CA enables detection and isolation of reactivated HIV-1+ cells with flow cytometry.

isolation from a screen. In future work, one may create fusion proteins that specifically manipulate or kill infected cells, e.g. by fusing a conditional Nb to a cellular toxin.

Studies of model organisms often take advantage of transgenic lines that express an exogenous protein in specific cell types (**Luo et al., 2008**). Driver molecules, such as transcription factors and site-specific recombinases, can respond to the introduction of DNA cassettes to enable the manipulation of gene expression in a cell type-specific manner. Here, we used GFP, which has no naturally known regulatory abilities, as a novel driver molecule. The GFP/dGBP1 binary system is thus analogous to the popular GAL4/UAS, Cre/loxP and Flp/FRT systems. Flp-DOG is immediately useful for studies in model organisms such as the mouse, by making use of existing transgenic GFP reporter lines (>1000 lines in the mouse) (**Chalfie, 2009**; **Gong et al., 2003**; **Heintz, 2004**; **Siegert et al., 2009**; **Tang et al., 2015**, **2013**) or virally labeled neural circuits (**Beier et al., 2011**; **DeFalco et al.,**

*2001*; *Ekstrand et al., 2014*; *Lo and Anderson, 2011*; *Schwarz et al., 2015*; *Wickersham et al., 2007*) for cell-specific manipulation studies. In addition, one can combine GFP and the popular Cre recombinase for intersectional Cre + Flp cell targeting studies (*Dymecki et al., 2010*; *Fenno et al., 2014*).

Destabilized nanobodies were originally developed with the desire to simplify the delivery of GFP-dependent reagents as well as to improve their performance. Indeed, we observed that the most direct output of the GFP/dGBP1 system – a dGBP1 fusion protein (dGBP1-TagBFP), could be stabilized by YFP with close to 100% efficiency amongst electroporated cells of the retina (*Figure 1—figure supplement 2B and C*). In comparison, the efficiency of direct reporter output using the T-DDOG and CRE-DOG systems was ~60% in similarly designed experiments, likely due to the need to co-deliver multiple reagents to the same cell (*Tang et al., 2015*, *2013*). Also, whereas the previous dimerizer-based systems were inhibited by excessive amounts of GFP, the activity of Flp-DOG continued to rise along with increasing GFP levels (*Figure 3—figure supplement 1*) (*Tang et al., 2015*, *2013*). These results highlight the improvements in the GFP/dGBP1 approach compared to previous approaches. However, there are several potential caveats of the GFP/dGBP1 system. First, although Flp-DOG is relatively easier to deliver to tissues as a single-component coding sequence, enhanced construct delivery may lead to enhanced background activity. Indeed, we found that it was necessary to remove the woodchuck hepatitis virus post-transcriptional regulatory element (WPRE) sequence from the rAAV expression cassette in order to avoid background Flp-DOG activity with rAAV infection. Second, tandemly repeated dGBP1s were fused to Flpo in order obtain tight GFP-dependent recombination. The requirement for two GFP molecules to stabilize a single Flp-DOG construct is predicted to reduce the sensitivity of Flp-DOG for lower GFP expression levels. However, in practice, Flp-DOG had adequate activity for the targeting of GFP cell types of the two different transgenic GFP lines tested. Nevertheless, these considerations suggest that one should establish an appropriate level of construct delivery, such as the amount of DNA plasmid or virus to deliver for optimal Flp-DOG performance.

Beyond fluorescent proteins, endogenous proteins should be usable as driver molecules to trigger sensor or effector activity. Endogenous proteins would enable one to selectively target specific cell types in wildtype animals for experimentation, without requiring any knowledge of cell type-specific promoters or creation of knock-in alleles. Such an approach would especially benefit studies of non-model organisms, with the only demand being a method to introduce genetic constructs, e.g. via viral vectors. As a possible caveat, the range of biological activities that could be driven by an endogenous protein may be limited by the protein's natural functions and/or sub-cellular localization. For example, it may be problematic to use a membrane-localized protein to trigger DNA recombination events in the nucleus. One may overcome this by choosing a dNb-fusion protein that, when stabilized, is biologically active in the same sub-cellular localization as that of the antigen. Another possibility is that an antigen-stabilized complex may travel from the site where stability is conferred to another sub-cellular locations to exert its function. These possibilities will likely be idiosyncratic to the fusion protein and antigen. An additional caveat is the effect that a dNb fusion protein might have on the targeted endogenous protein. There could be cases where antigen-Nb interactions lead to less antigen activity. Reduction in activity will depend on several variables. The ratio of Nb fusion to antigen, the binding site of the Nb fusion, and perhaps the particular Nb fusion structure, all have the potential to create changes in antigen activity. Whether or not a reduction in cellular function will be effected will depend upon the sensitivity of the cell to the level of antigen activity. In most heterozygous loss of function mutations in mice and humans, a phenotype is not noted. Although we cannot predict the frequency with which Nb fusions will result in a phenotype, we believe that the method is strong enough to encourage its continued development and application.

## Optimization of antigen-dependent sensors and effectors

Although dGBP1-TagBFP showed virtually no background signal, fusions of dGBP1 to some fusion partners gave significant background signals. Background signals could be addressed by simply increasing the number of dNbs fused to the protein partner. Additional engineering efforts could further reduce background activity of particular fusion constructs. For example, background activity of dGBP1-Cre could be further controlled by fusion to an ERT2 domain to create small-molecule dependency (*Feil et al., 1997*). Lastly, one could perform additional screens to isolate novel

destabilizing mutant combinations that enhance the antigen-specificity of a wider variety of sensor/ effector fusion partners. Such an approach could help eliminate the background fluorescent aggregates seen with some dNb-fluorescent protein fusions.

Additional improvements may be made to enhance the response of dNb fusion constructs to antigen co-expression. First, the activity of a protein might be affected by fusion to the Nb. For example, Cre activity was reduced upon binding of GFP to GBP1-Cre, possibly as a result of steric hindrance. Second, the sensitivity of the dNb fusion construct might be sub-optimal. Although the proportion of CA+ cells detected by the dNb sensor was approximately 1/3 to 1/2 of that detected by a mouse monoclonal antibody, the efficiency of antigen detection may be improved by optimization of the dNb fusion construct or of the gene delivery protocol. Possible optimization steps may include exploring different fusion orientations, linker lengths or linker compositions.

## Possible applications with dNb-based sensors and effectors

The fusion of protein binders to fluorescent proteins enables visualization of antigen localization in living cells. Optimal signal-to-noise detection requires that the fluorescent fusion proteins be strictly co-localized with the antigen. This may not occur if the number of fluorescent fusion proteins exceeds the number of target antigens. One could address this by designing a transcriptional feedback mechanism to control the level of a fluorescent fusion protein (*Gross et al., 2013*). This method requires that the antigen be localized outside of the nucleus. The use of dNb fluorescent protein fusions is not limited by this requirement, as the mechanism for background reduction involves protein degradation rather than transcriptional feedback. Indeed, we found that a dNb-TagBFP construct became strictly localized to the nucleus upon co-expression with its NLS-tagged antigen (data not shown).

The finding that one can fuse an effector protein to two dNbs led to the development of a strategy wherein two different antigens bound to distinct dNbs could stabilize the effector protein, Flpo. With further development, dual antigen dependence may enable one to precisely target specific cell populations in ways similar to established intersectional strategies, but using proteins that do not necessarily have any defined regulatory abilities (*Dymecki et al., 2010*; *Luo et al., 2008*).

Here, we demonstrated the ability to integrate dNb with CRISPR/Cas technology to perform genome editing selectively in antigen-expressing cells. A concern with the expression of Cas9 and gRNA in cells is that there is non-specific genome editing, and several methods are being developed to address this problem (*Hsu et al., 2014*; *Sander and Joung, 2014*). The strategy developed here, wherein dNb-Cas9 activity is suppressed until antigen can stabilize the fusion protein, offers a novel strategy to restrict the cell populations that might suffer from off-targeting events.

## Generation of additional destabilized binder systems with screens and/ or mutation transfer

The screening strategy described here should enable the creation of additional protein-responsive reagents useful for control of fusion protein activity in specific cell populations. A key feature of our screen is the use of rAAVs to deliver the antigen to cells. Virtually all MMLV-infected cells in culture can be super-infected by rAAV, and the cells remain viable for subsequent culture expansion and FACS. In principle, this screening strategy can be extended to generate a diversity of protein-responsive, destabilized binders based on the Nb scaffold or other protein scaffolds (*Wurch et al., 2012*; *Helma et al., 2015*).

Over the past 20 years, ~100 crystal structures featuring Nb-antigen complexes have been solved. We leveraged this resource to establish a phylogenetic and structural basis for transferring destabilizing mutations across Nbs. All successfully modified Nbs were derived from camelid species different from that of GBP1, demonstrating the broad transferability of the mutations discovered here. As one would expect, dNb-TagBFPs generated by mutation transfer showed a spectrum of fluorescence fold-change in response to antigen co-expression. This is likely due to multiple factors, including variable Nb affinity for antigen, variable antigen stability, and variable Nb stability even before destabilization. In addition, Nbs might have variable tolerance to the destabilization mutations tested. Thus, although the high percentage of successful mutation transfers indicate that the strategy is generally applicable, it would be beneficial to derive novel combinations of destabilizing mutations that are even better tolerated across Nbs. Lastly, although dNb generation may be limited by the availability

of Nbs isolated from immunized animals, additional Nbs and dNbs may be isolated from in vitro screening technologies that are constantly being improved upon.

Beyond Nbs, multiple classes of artificially derived binding proteins that are amenable to expression in living cells are being developed for antigen-recognition (*Helma et al., 2015*; *Wurch et al., 2012*). As epitope-specific binders are typically generated by varying loops or surfaces on a common structural scaffold, it should be possible to generate epitope-responsive properties by incorporating a common set of mutations onto conserved and non-epitope binding regions of the scaffold. Future developments building upon this work should expand our ability to rapidly generate sensors and effectors against a diversity of intracellular epitopes, for cell- or antigen-specific applications in biology and medicine.

# Materials and methods

## Animals

The Institutional Animal Care and Use Committee at Harvard University approved all animal experiments. Timed pregnant CD1 (Charles River Breeding Laboratories, Boston, MA) were used for electroporation experiments. Tg(CRX-GFP) (*Samson et al., 2009*) and Tg(GAD67-GFP) (*Tamamaki et al., 2003*) were kept on a C57/BL6J background.

## Miscellaneous plasmids

pCAG-GFP (Addgene plasmid 11150) (*Matsuda and Cepko, 2004*). pCAG-YFP (Addgene plasmid 11180) (*Matsuda and Cepko, 2004*). pCAG-DsRed (Addgene plasmid 11151) (*Matsuda and Cepko, 2004*). pRho-GFP-IRES-AP (referred to as Rho-GFP) (*Emerson and Cepko, 2011*). pCAG-nlacZ (Cepko lab, Harvard Medical School) pCAGEN (Addgene plasmid 11160) (*Matsuda and Cepko, 2004*). pCALNL-DsRed (Addgene plasmid 13769) (*Matsuda and Cepko, 2004*). pCAFNF-DsRed (Addgene plasmid 13771). (*Matsuda and Cepko, 2004*). pCALNL-luc2 (*Tang et al., 2015*). pRL-TK (#E2241; Promega, Madison, WI).

## Antibodies

Antibodies used were rabbit anti-TagRFP (also targets TagBFP; 1:5,000 dilution for immunoblot, 1:1,000 for immunohistochemistry) (AB233; Evrogen, Moscow, Russia), mouse anti-βgal (1:50 for immunoblot) (40-1a supernatant; Developmental Studies Hybridoma Bank, University of Iowa), chicken anti-βgal (1:1,000 for immunohistochemistry) (ab9361;Abcam, Cambridge, MA), rabbit-anti-GFP (1:500 for immunohistochemistry) (A-6455; Invitrogen, Carlsbad, CA), rabbit anti-GAPDH (1:10,000 for immunoblot) (A300-641A; Bethyl Laboratories, Inc., Montgomery, TX), mouse anti-FLAG M2 (1:1,000 for immunoblot) (F1804; Sigma-Aldrich, St. Louis, MO), mouse anti-KC57-RD1 (5ul per 1 million cells) (6604667; Beckman Coulter, Danvers, MA). Secondary antibodies used were goat anti-chicken Alexa Fluor 647 (1:500 of 50% glycerol stock) (102371; Jackson ImmunoResearch Laboratories Inc., West Grove, PA), goat anti-rabbit DyLight 649 (1:500 of 50% glycerol stock) (111-495-144; Jackson ImmunoResearch Laboratories Inc.), anti-rabbit or anti-mouse IgG-Horseradish Peroxidase (GE Healthcare, Little Chalfront, UK).

## Cell lines

293T cells (Cepko lab stock). ACH-2 (NIH AIDS Reagent Program, Germantown, MD). CEM (NIH AIDS Reagent Program). HIV-1 integration sites in ACH-2 cell line were authenticated by integration site analysis of HIV-1 genome, confirming the major integration site is on chromosome 7. Using immunostaining followed by flow cytometry analysis, HIV-1 CA protein was confirmed to be present in ACH-2 cells, and to be absent in CEM cells.

## Screen for dGBP1

### Generation of mutagenized GBP1 Library

GBP1 and flanking vector sequences were PCR amplified from the pBMN-GBP1-TagBFP vector (described below). A SphI site was inserted between the GBP1 and TagBFP sequences, creating an AC amino-acid linker. For mutagenesis, GBP1 was amplified by PCR to add 5' and 3' overhangs corresponding to vector sequence flanking GBP1 in the desired pBMN-GBP1-TagBFP construct. The

amplified products were then randomly mutagenized using primers targeting the overhang sequences that flank GBP1. The primers were: Forward primer 5'GACCATCCTCTAGACTGCCGGA TCCGCCACC-3'Reverse primer 5'-TGTTCTCCTTAATCAGCTCGCTCATGCATGC-3'.The Gene-Morph II Random Mutagenesis Kit was used to introduce balanced mutation rates for different nucleotides, and at high, medium or low mutation frequency (Agilent, Lexington, MA). Mutagenized GBP1 DNA was inserted into a BamHI/SphI-digested, pBMN-GBP1-TagBFP vector by Gibson Assembly. This created an in-frame fusion with TagBFP bridged by the SphI linker. Transformed DH5α were grown overnight and harvested for DNA purification using Maxiprep kits (QIAGEN, Hilden, Germany). An aliquot of each culture that grew well was plated, and GBP1 inserts were sequenced. Between 9,000–160,000 colonies were produced per library preparation, with 85% of sequenced colonies carrying a unique combination of GBP1 mutations. A library using the medium mutation rate was used to generate an MMLV library using VSV-G for an envelope (*Yee et al., 1994*).

## Selection of candidate GBP mutants

293T cells infected with the MMLV library encoding GBP1 mutants were sorted by FACS for presence of red fluorescence (from IRES t-HcRed) and for absence of blue fluorescence (from putative destabilized TagBFP). As a control, 293T cells infected with MMLV encoding unmodified GBP1 was used to establish gating for TagBFP expression. Sorted cells were plated and allowed to expand in culture. Cells were then infected with rAAV-EF1a-GFP 2/8 virus (*Tang et al., 2015*). 24 hr later, cells were sorted for red fluorescence and high blue fluorescence. Sorted cells were then seeded into T25 flasks and allowed to grow to confluence. Next, cellular DNA was extracted using the DNeasy kit (QIAGEN). GBP1 variants were PCR amplified by Phusion polymerase (New England Biolabs, Ipswich, MA), using primer targeting vector sequences flanking GBP1. PCR products were inserted into the pBMN vector with Gibson Assembly. One hundred bacterial colonies were picked and sequenced. Plasmids from clones were individually transfected into 96 well plates of 293T cells with or without CAG-YFP to assay for TagBFP fluorescence. Almost all isolated GBP1-TagBFP variants showed YFP-dependent fluorescence, but many had either TagBFP aggregation or high background fluorescence in the absence of YFP. The only clone that showed a complete lack of TagBFP fusion fluorescence when YFP was absent was named dGBP1 (A25V, E63V, S73R, C/S98Y, Q109H, S117F).

## General strategy for cloning Nbs and antigens into pCAG vector

All antigen and Nb protein sequences, except YFP, were acquired from Protein Data Bank (PDB). Protein sequences were backtranslated into DNA sequences, using codons optimized for *Mus musculus*. The list of tested Nbs and their antigens are listed in *Table 1*. In general, an antigen sequence was synthesized as gBlock fragments, which were inserted into an EcoRI/NotI digested pCAG vector via Gibson Assembly, giving pCAG-antigen plasmids used for co-expression of antigen in cells. In general, Nb sequences were synthesized as gBlock fragments, and individually inserted into an EcoRI/SphI digested pCAG-TagBFP vector via Gibson Assembly, giving pCAG-Nb-TagBFP plasmids. To destabilize Nbs, mutations were introduced into residue positions that aligned with the dGBP1 mutation positions. Equivalent residues were easy to identify since surrounding amino acid sequences were highly conserved. For 6mut combo, the dGBP1 mutations were A25V, E63V, S73R, C/S98Y, Q109H and S117F. For 3maj combo, the dGBP1 mutations were S73R, S98Y and S117F. Note that C/S98Y in GBP1 was originally a cysteine, but was mutated to serine in earlier studies to avoid complications with disulfide bond formation.

## Construction of selected DNA constructs

*pBMN-GBP1-TagBFP* - A GBP1-TagBFP construct was inserted into a BamHI/NotI digested pBMN DHFR(DD)-YFP (a gift from Thomas Wandless; Addgene plasmid # 29325) (*Iwamoto et al., 2010*), replacing the DHFR(DD)-YFP insert and generating pBMN-GBP1-TagBFP. This became the host vector for mutagenized GBP1 inserts.

 *pBMN-dGBP1-Cre and pBMN-dGBP1-Flpo* –pBMN-dGBP1-TagBFP were digested with SphI/SalI, liberating TagBFP as well as the IRES-t-HcRed element. PCR-amplified Cre and Flpo fragments were then inserted into the digested vector via Gibson Assembly.

*pBMN-GBP1-Cre and pBMN-GBP1-Flpo* – PCR fragments of GBP1 were inserted into BspEI/SphI digested pBMN-dGBP1-Cre and pBMN-dGBP1-Flpo by Gibson Assembly, resulting in pBMN-GBP1-Cre and pBMN-GBP1-Flpo, respectively. dGBP1 sequence was removed by BspEI/SphI digest.

*pBMN-dGBP1x2-Cre, pBMN-dGBP1-GBP1-Cre, pBMN-dGBP1x2-Flpo, pBMN-dGBP1-GBP1-Flpo* – pBMN-dGBP1-Cre or –Flpo plasmids were digested with SphI. A gBlock fragment encoding a codon modified dGBP1 was inserted into this site via Gibson Assembly, generating pBMN-dGBP1x2-Cre or –Flpo. Using a GBP1 gBlock fragment instead of dGBP1 gave pBMN-dGBP1-GBP1-Cre or –Flpo.

*pCAFNF-luc2* – An EcoRI-Kozak-luc2-NotI DNA fragment liberated from pCALNL-luc2 (*Tang et al., 2015*) was sub-cloned into EcoRI/NotI digested pCAFNF-DsRed vector, giving pCAFNF-luc2.

*pCAG-dGBP1-TagBFP* – Using PCR, an AgeI-Kozak-dGBP1-TagBFP-NotI was generated from pBMN-dGBP1-TagBFP. This fragment was sub-cloned into AgeI/NotI digested pCAG-GFP, giving pCAG-dGBP1-TagBFP and removing GFP from the construct.

*pCAG-dGBP1-TagBFP-FLAG* – A gBlock fragment encoding Kozak-TagBFP-FLAG was inserted into SphI/NotI digested pCAG-dGBP1-TagBFP via Gibson Assembly, giving pCAG-dGBP1-TagBFP-FLAG and removing untagged TagBFP from the construct.

*pCAG-YFP-FLAG* – A gBlock fragment encoding Kozak-YFP-FLAG was inserted into EcoRI/NotI digested pCAG-αCA-dNb$^{6mut}$-TagBFP, giving pCAG-YFP-FLAG and removing αCA dNb$^{6mut}$-TagBFP from the construct.

*pCAG-dGBP1-mCherry* – PCR amplified mCherry was inserted into a SphI/NotI digested pCAG-dGBP1-TagBFP vector, resulting in replacement of TagBFP with mCherry. The vector became pCAG-dGBP1-mCherry.

*pCAG-GBP1-mCherry* – A gBlock fragment encoding GBP1 was inserted into a EcoRI/SphI digested pCAG-dGBP1-mCherry vector, resulting in replacement of dGBP1 with GBP1. The vector became pCAG-GBP1-mCherry.

*pCAG-αCA-Nb-TagBFP, pCAG-αDHFR-Nb-TagBFP, pCAG-αCA-dNb$^{6mut}$ -TagBFP and pCAG-αCA-dNb$^{3maj}$-TagBFP* –A gBlock fragment carrying either the αCA Nb or αDHFR Nb coding sequence was inserted into an EcoRI/SphI digested pCAG-TagBFP vector via Gibson Assembly, resulting in pCAG-αCA-Nb-TagBFP or

pCAG-αDHFR-Nb-TagBFP. gBlocks carrying these mutations in the respective Nbs were introduced into the EcoRI/SphI digested pCAG-TagBFP vector via Gibson Assembly, giving either pCAG-αCA-dNb$^{6mut}$-TagBFP or pCAG-αDHFR-dNb$^{3maj}$-TagBFP.

*pCAG-dGC-Flpo and pCAG-dGD-Flpo* – A gBlock fragment carrying either αCA-dNb$^{6mut}$ or αDHFR-dNb$^{3maj}$ coding sequence were inserted into SphI digested pCAG-dGBP1-Flpo vector (Tang, J.C.Y., Cepko lab) via Gibson Assembly, giving either pCAG-dGC-Flpo or pCAG-dGD-Flpo, respectively.

*pCAG-dGBP1x2-Flpo* – An AgeI-Kozak-dGBP1x2-Flpo-NotI fragment was generated by PCR using pBMN-dGBP1x2-Flpo as a template. This fragment was sub-cloned into AgeI/NotI-digested pCAG vector, giving pCAG-dGBP1x2-Flpo.

*pCAG-αCA-dNb$^{6mut}$x2-Flpo* – Two gBlock fragments, together encoding αCA-dNb$^{6mut}$x2, was inserted into EcoRI/SphI-digested pCAG-dGBP1x2-Flpo, giving pCAG-αCA-dNb$^{6mut}$x2-Flpo and replacing dGBP1x2 from the construct.

*pCAG-αCA-dNb$^{6mut}$-TagRFP* – A gBlock fragment carrying the TagRFP coding sequence was inserted into SphI/NotI digested pCAG-αCA-dNb$^{6mut}$-TagBFP via Gibson Assembly, giving pCAG-αCA-dNb$^{6mut}$-TagRFP and removing TagBFP from the construct.

*pAAV-EF1α-dGBP1x2-Flpo-NW*- A BamHI-Kozak-dGBP1x2-Flpo-EcoRI PCR fragment was inserted into BamHI/EcoRI digested pAAV-EF1α-N-CretrcintG (*Tang et al., 2015*), giving pAAV-EF1α-dGBP1x2-Flpo. The WPRE element was subsequently removed from this plasmid via EcoRV/AfeI digest and re-ligation, giving pAAV-EF1α-dGBP1x2-Flpo-NW.

*pAAV-CAG-FLEX$^{FRT}$-ChR2(H134R)-mCherry*- A Chr2(H134R)-mCherry DNA fragment was digested with NheI and inserted into NheI digested pAAV-CAG-FRTed-SynGFPreverse-WPRE (*Pivetta et al., 2014*) (a gift from Sylvia Arber) (*Pivetta et al., 2014*). A clone with ChR2(H134R)-mCherry inserted in the reverse direction relative to CAG promoter were chosen, giving pAAV-FLEX$^{FRT}$-ChR2(H134R)-mCherry.

*pCAG-C-CA and pCAG-DHFR*– A gBlock fragment carrying either the HIV-1 C-CA coding sequence (encoding residue 278–352 of HIV-1 gag polyprotein) or *E.coli* DHFR coding sequence was inserted into EcoRI/NotI digested pCAG-GFP vector via Gibson Assembly; C-CA or DHFR replaced GFP in the cassette.

## Cell culture data

For sample size, we chose to reproduce all our results in at least 3 independent experiments (equal to at least 3 biological replicates, or transfected wells). We consider this to be a sufficient sample size for demonstrating reproducibility of our findings. For statistical analysis, we chose to increase our independent experiments (equal to biological replicates) to 8.

## Cell culture and transfection

293T cells were seeded onto 24 or 96 well plates and used for transfection when the cells reached between 60–95% confluency, usually 1–2 days later. Transfections were achieved with polyethylenei-mine (PEI) at a 1:4 DNA mass:PEI volume ratio. PEI stock was 1 mg/ml. A total of between 100 and 400 ng of DNA were transfected into single wells of 96 well plates for fluorescence analysis of desta-bilized mutants. Approximately 70 ng total DNA were transfected into single wells of 96 well plates for luciferase analysis. Approximately 400 to 520 ng of total DNA were transfected into single wells of 24 well plates for fluorescence imaging and western blot analysis.

## Cell culture fluorescence imaging experiments

### General information

All cell culture images were acquired on a Leica DMI3000B microscope (Leica, Wetzlar, Germany), using a 5x, 10x or 20x objective. pCAG-YFP was used in place of pCAG-GFP to induce dGBP1-TagBFP stability in order to avoid fluorescence bleedthrough from brightly fluorescent GFP signals into the TagBFP channel.

### YFP-specificity of dGBP1-TagBFP fluorescence

293T cells seeded in 96 well plates were transfected with 200 ng pBMN-GBP1-TagBFP, pBMN-dGBP1-TagBFP, or pCAGEN along with 27.5 ng of pCAG-YFP or pCAGEN. Cells were imaged for fluorescence 2 days post-transfection.

### Antigen-specific DsRed activation with Flpo constructs

All transfection conditions were adjusted to 180 ng total DNA and transfected into 293T cells in 96 well plates. 50 ng of pBMN-based plasmids encoding dGBP1-Cre, dGBP1-Flpo, dGBP1x2-Cre, dGBP1x2-Flpo, dGBP1-GBP1-Cre, or dGBP1-GBP1-Flpo were used. 100 ng of pCALNL-DsRed or pCAFNF-DsRed were used as reporter readouts of Cre or Flpo recombination, respectively. 30 ng of pCAG-GFP or pCAG-YFP were used for antigen co-expression conditions, whereas the same amount of pCAGEN replaces GFP or YFP plasmids in negative control conditions. Cells transfected with Cre- or Flpo- fusion constructs were imaged for 22 or 50 hr post-transfection, respectively. Flpo- fusion constructs were less active and thus required a longer incubation time to detect signal.

## Mapping of dGBP1 mutations

### Transfection

293T cells seeded in 96 well plates were transfected with 75 ng pCAG-driven GBP1 variant con-structs along with 75 ng of either pCAG-GFP or pCAGEN. Cells were imaged 17 hr post-transfection.

### Analysis

To map the effects of individual dGBP1 mutations on protein stability, we scored by eye the fluores-cent intensity and solubility of mCherry tagged with various GBP1 variants. We use a semi-quantita-tive approach to score mCherry intensity, based on a six point scale ranging from 0 to 3, with 0.5 point increments. For solubility scores, we used a 4 point score ranging from "soluble" (mCherry dif-fusely distributed in cytoplasm), "soluble, some aggregate" (mostly diffuse mCherry expression but

some instances of mCherry aggregation), "soluble/aggregate" (mixture of diffuse mCherry and aggregating mCherry), and "aggregate" (strongly aggregating mCherry). As a reference point for both intensity and solubility scores, we compared the intensity of variants to that of either GBP1-mCherry and/or dGBP1-mCherry controls. Scores were assessed across replicates and in independent experiments.

## Mapping of dGBP1 mutations across nanobody-antigen interfaces

We exhaustively searched the Protein Data Bank (PDB) and, at the time of analysis, identified 77 unique camelid single-chain antibody fragments (VHH or VH, here collectively referred to as nanobodies (Nbs)) that have been co-crystallized with their respective antigens. We removed one structure (PDB ID 3J6A) from the analysis because it was a low-resolution structure produced by cryo-electron microscopy. We used the PDBePISA online server tool (http://www.ebi.ac.uk/pdbe/pisa/) to evaluate whether residue positions equivalent to those of dGBP1 mutations are located outside of antigen-Nb interfaces across different crystallized complexes. PDBePISA produces an analysis of buried surface area (BSA), defined as the solvent-accessible surface area of the corresponding residue that is buried upon interface formation, in $\text{Å}^2$. We considered an Nb residue to be in the interface with the antigen if its BSA value is above 0 $\text{Å}^2$. We confirmed that this metric is a reliable indicator of an Nb residue's proximity to the antigen by examining all structures by eye using tools such as PyMol. In a few cases the Nb bound to more than one antigen. We took this into consideration by analyzing the interfaces formed between an Nb and each of the two antigens. We used protein alignment to determine the residue positions corresponding to the mutations located in dGBP1. The same 76 Nbs were used to determine the extent of GBP1 residue conservation across Nbs.

Analysis across 76 unique Nb-antigen interfaces, and a total of 102 uniquely crystallized interfaces, indicated that Nb positions corresponding to those of dGBP1 A25V, S73R, S98Y and S117F were universally located outside of all Nb-antigen interfaces (*Figure 2—figure supplement 3B*). Nb positions corresponding to dGBP1 Q109H fell outside of 99%, or 75 of 76 unique Nb-antigen interfaces. Positions equivalent to dGBP1 E63V were directly found in 22%, or 17 of 76 unique Nb-antigen interfaces, and in close proximity to the interface in 9%, or 7 of 76 of the cases.

16 identical or highly similar Nb-antigen complexes had been crystallized under similar or differing conditions, allowing us to validate the mutation mapping results by comparing across identical or related crystal structures. There was agreement in mutation mapping results between redundant or similar crystal structures for 94%, or 16 of 17 unique Nb-antigen interfaces. The lone exception concerned a unique Nb (PDB ID 4KRM and 4KRL). The Q109H equivalent position in the Nb of structure 4KRL was identified to be at the interface. This discrepancy may be explained by the fact that the two structures were crystallized under very different pH conditions.

## Selection of Nbs and antigens for mutation transfer experiment

For the mutation transfer experiment, we tested 18 Nbs that, when fused to TagBFP, showed strong blue fluorescence and soluble, diffuse localization in 293T cells. Some available nanobodies were too problematic to be included for analysis because they recognized problematic antigens such as the Ricin toxin. Nevertheless, Nbs that bound to proteins originating from both intracellular and extracellular locations were selected. We used a pCAG expression vector to express antigen deposited in PDB for each crystal structure. During analysis, we noticed a strong correlation between dNbs that failed to be stabilized by antigen (<2-fold TagBFP fluorescence induction by antigen) and the use of extracellular epitopes. dNbs targeting extracellular epitopes were thus excluded from evaluation of mutation transfer generality.

To determine whether the antigen used for TagBFP stabilization assays derived from intracellular or extracellular proteins, we studied the annotation and literature reports of each antigen's cellular localization.

## Nb destabilizing mutation transfer experiments

### Transfection

293T cells seeded in 96 well plates were transfected with the following plasmid mix: 50 ng of CAG-driven Nb-TagBFP or dNb-TagBFP fusion constructs, 75 ng of CAG-driven DsRed (CAG-DsRed), and either 150 ng of CAG-driven antigen corresponding to the Nb of interest, or an equivalent amount

of empty vector (pCAGEN) in negative control conditions. The DNA mix was transfected with PEI at a ratio of 1:4 (DNA µg:PEI µl) ratio. 16–24 hr post-transfection, DsRed and TagBFP fluorescence images were acquired on a Leica DMI3000B microscope, using a 20x objective. DsRed served as a marker of transfected cells, and guided the imaging of TagBFP fluorescence regardless of condition.

## Image analysis

Cell culture images were processed on ImageJ. DsRed and TagBFP images were converted to 8-bit. DsRed images were adjusted with the threshold function, converted to binary images, and processed with the "fill holes" and "watershed" functions to create individual regions of interests (ROI) that represent single cells. Processed DsRed binary images were used to guide measurement of mean pixel intensity of TagBFP images in the regions of interests (ROI). Measurements were only made on ROIs that had areas larger than 0.01 inches$^2$. To measure background fluorescence of TagBFP images, 5–15 squares were drawn in areas devoid of cells and collectively measured for a single mean pixel intensity value per image. This background value was subtracted from individual pixel intensity measurements from each ROI. This background subtraction approach occasionally produced negative values. Negative values were set to zero to enable scaling of data. This manipulation did not affect the representation of the data, since we used the median to represent center of spread and all median values were found to be above zero. Data from each experiment were divided by the median value of the corresponding "unmodified Nb, no antigen condition", and multiplied by 100 to get a percentage of wildtype fluorescence level. Fold induction was obtained by dividing the median normalized TagBFP fluorescence reading in the "with antigen" condition by the equivalent value in the "without antigen" condition. TagBFP fluorescence measurements and fold induction values were analyzed from at least 3 micrographs, taken from 3 independent experiments. Heat map showing normalized TagBFP fluorescence values was generated in Excel. For each condition, all three replicate values are shown in a series of 3 horizontal colored bars. The color gradient was chosen to emphasize changes between 0 and 40%, because most dNb-TagBFP constructs gave values in this range. Graphs were plotted as the median TagBFP pixel intensity. Values were plotted in Prism (Graphpad, La Jolla, CA).

## Luciferase assay experiments

### General information

In all experiments, 20 ng CALNL-luc2 or CAFNF-luc2 and 3 ng pRL-TK were included in transfection mixture delivered to 293T cells seeded in 96 well plates. Plasmids encoding CAG-driven XFP and dNb fusion constructs were transfected at amounts adjusted for their molarity. pCAGEN was added to adjust the total DNA amount to approximately 70 ng. Cells were harvested at the appropriate time for Dual-luciferase assay (Promega) according to manufacturer's instructions. Lysates were pipetted into 96-well plates and read in a Spectra Max Paradigm plate reader (Molecular Devices, Sunnyvale, CA). The linear range of detection for the plate reader was determined with serial dilutions of QuantiLum recombinant luciferase (Promega). Transfection amounts were then adjusted to give readings within the linear range of detection for the instrument. All transfection conditions were independently repeated at least 3 times and were assayed in one to three replicates in terms of transfection wells (biological replicates) and/or plate reader well (technical replicates). Luciferase readings were processed similarly to previous studies (*Tang et al., 2013*, *2015*). Fold induction was determined by dividing the mean normalized luciferase activity of the "with antigen" condition by that of the 'without antigen' condition.

### Antigen-specificity of Flpo constructs

All transfection conditions were adjusted to include a total of 180 ng DNA. 45–50 ng of pBMN-based plasmids encoding GBP1-Cre, GBP1-Flpo, dGBP1-Cre, dGBP1-Flpo, dGBP1x2-Cre or dGBP1x2-Flpo were tested for Cre or Flpo-dependent recombination of luciferase reporter. 20 ng of pCAG-GFP was used for co-expression conditions, whereas the same amount of pCAGEN replaced GFP plasmids in negative control conditions. Cre or Flpo transfected cells were harvested at 15 or 36 hr post-transfection, respectively. Readings were normalized against a specific condition such that the background reporter activity gave a value of 1.

## Flpo recombination dependent on two different antigens

A total of 57 ng DNA was transfected. 4 ng of pCAG-dGD-Flpo or pCAG-dGC-Flpo were tested for Flp-dependent recombination, along with 15 ng pCAG-GFP and/or 15 ng pCAG-C-CA or pCAG-DHFR. pCAGEN was used as a filler for antigen-expressing plasmids in all cases. Cells were harvested 16 hr post-transfection.

## Dose-dependency of Flpo constructs

All transfection conditions were adjusted to 178 ng total DNA. 5 ng of pCAG-dGBP1x2-Flpo or pCAG-αCA-dNb$^{6mut}$x2-Flpo were used. CAG-DsRed was used as a filler plasmid to substitute for the antigen-expressing plasmid. Antigen-expressing plasmids were tested at a range from 150 ng to 0.6 ng. pCAG-GFP and pCAG-C-CA were used to test for antigen-dependency of dGBP1x2-Flpo and αCA-dNb$^{6mut}$x2-Flpo, respectively. CAG-DsRed was used as a filler plasmid to substitute for the antigen-expressing plasmid. Transfected cells were harvested for luciferase assay at 24 hr post-transfection. Normalized luciferase activity of a transfection condition dropping out antigen-expressing plasmid and Flpo-expressing plasmid was subtracted from readings of all other conditions in the same experiment.

## Western blot experiments

### General information

293T cells were seeded onto 24 well plates and transfected using PEI. pCAG-nlacZ was used as a transfection loading marker. Transfected 293T cells were lysed in 6xSDS PAGE loading buffer (350 mM Tris-HCl (pH = 8), 30% glycerol, 10% SDS, 600 mM DTT, 0.01% Bromophenol Blue), heated to 95°C for 5 min, and stored at −20°C until used for western blot analysis. When necessary, transferred blots were cut into two pieces for blotting with different antibodies.

### Stabilization of dNb-TagBFP or of antigen

To demonstrate the ability of YFP to stabilize dGBP1, 400 ng pBMN-GBP1-TagBFP or pBMN-dGBP1-TagBFP were transfected into 293T cells along with 55 ng of pCAG-YFP or pCAGEN. 60 ng of pCAG-nlacZ was included in all conditions as a transfection marker. Cells were harvested for western blot 2 days post-transfection. To test the ability of C-CA to stabilize destabilized αCA-dNb$^{6mut}$, 100 ng pCAG-αCA-Nb-TagBFP or pCAG-αCA-dNb$^{6mut}$-TagBFP were transfected into 293T cells along with 125 ng pCAG-C-CA or pCAGEN. 200 ng pCAG-nlacZ was also included in the mix. To test the ability of DHFR to stabilize destabilized αDHFR-dNb$^{3maj}$, 100 ng pCAG-αDHFR-Nb-TagBFP or pCAG-αDHFR-dNb$^{3maj}$-TagBFP were transfected into 293T cells along with 375 ng pCAG-DHFR or pCAGEN. 50 ng pCAG-nlacZ were also included in the mix. Cells transfected with C-CA or DHFR test constructs were harvested 16 hr post-transfection. To test the ability of dGBP1-TagBFP to stabilize YFP, 150 ng pCAG-dGBP1-TagBFP or pCAG-dCA$^{6mut}$-TagBFP were transfected into 293T cells along with 100 ng pCAG-YFP and 300 ng pCAG-nlacZ. Cells were harvested 1 day post-transfection.

### Proteasome inhibition experiment

To test whether dGBP1 degradation was dependent on the UPS, 293T cells were transfected with 400 ng pBMN-GBP1-TagBFP or pBMN-dGBP1-TagBFP. 60 ng pCAG-nlacZ were added as a transfection marker, while 60 ng pCAGEN were added as a filler plasmid. 20–24 hr post-transfection. Cells were treated with 10 µM MG132 (C2211; Sigma Aldrich) or 10 µM Bortezomib (sc-217785, Santa Cruz Biotech, Dallas, TX) for 20 hr before harvesting for western blot. To test whether the degradation of αCA-dNb$^{6mut}$-TagBFP was dependent on ubiquitin proteasome action, 293T cells were transfected with CAG-driven plasmids encoding 150 ng αCA-Nb-TagBFP or αCA-dNb$^{6mut}$-TagBFP, along with 300 ng pCAG-nlacZ. 10 hr post-transfection, cells were treated with 10 µM MG132 for 6 hr. Cells were then harvested for western blot.

### Dose experiment

To assay for dose dependency of dGBP1-TagBFP-FLAG on YFP, a total of 750 ng of DNA were transfected into 293T cells. All transfections conditions included 200 ng pCAG-nlacZ, and 50 ng pCAG-dGBP1-TagBFP-FLAG. pCAG-YFP-FLAG was used in the range of 1.8 ng to 500 ng. In one condition, 500 ng pCAG-YFP was used in place of pCAG-YFP-FLAG. pCAG-DsRed was used as a

filler plasmid to adjust for pCAG-YFP-FLAG removal, up to 500 ng. Transfected cells were harvested 24 hr post-transfection for western blot. Mouse-Anti-FLAG was used to detect the presence of both dGBP1-TagBFP-FLAG (41 kDa) and YFP-FLAG (28 kDa). dGBP1-TagBFP-FLAG stabilized by untagged YFP helped confirm that the 28 kDa protein was YFP-FLAG rather than a degradation product of dGBP1-TagBFP-FLAG.

## Densitometry analysis

To quantify western blot bands, we followed the protocol laid out in: http://www1.med.umn.edu/starrlab/prod/groups/med/@pub/@med/@starrlab/documents/content/med_content_370494.html. We quantified the anti-TagBFP immunopositive bands (~39.5 kDa, migrating close to 38 kDa marker band), anti-YFP immunopositive bands (~27 kDa), anti-βgal immunopositive bands (~120 kDa, migrating between 98 and 198 kDa marker bands) as well as background signal bands in empty lanes to arrive at the adjusted relative density for anti-TagBFP bands. Two-tailed Student's t-test assuming unequal variance was used for testing of statistical significance. $P < 0.05$ is judged as statistically significant.

## In vivo data

In all in vivo experiments, biological replicates are defined in terms of cells, retinas or animals. Technical replicates are defined in terms of whole brain sections. We consider 3 biological replicates to be a sufficient sample size for demonstrating reproducibility of our findings. As an exception, we had 2 biological replicate for injection of green fluorescent beads/rAAV mix into GFP and wildtype mouse brains (*Figure 5—figure supplement 2*). However, we deemed this sufficient as the results basically replicated our findings in an equivalent experiment using a slightly different injection mix (*Figure 6*). For statistical analysis, we used data that consisted of 7–21 cells.

## In vivo electroporation experiments

### General information

Postnatal day 0 (P0) or P2 mouse pups were microinjected with plasmids into their subretinal space and subjected to electroporation (*Matsuda and Cepko, 2004*).

### Testing of dGBP1- TagBFP in vivo

1.33 µg/µl pCAG-dGBP1-TagBFP were injected into CD1 mice along with 1.33 µg/µl of pCAG-DsRed, pCAG-GFP or pRho-GFP. Electroporated CD1 retinas were harvested at P14, immunostained for anti-TagBFP antibodies in the far-red channel, and imaged by confocal microscopy.

### Testing of αCA-dNb⁶ᵐᵘᵗx2-Flpo in vivo

0.33 µg/µl pCAG-αCA-dNb$^{6mut}$x2-Flpo, 0.42 µg/µl pCAFNF-DsRed and 0.42 µg/µl pCAG-nlacZ were injected into CD1 retinas along with 0.5 µg/µl of either pCAG-C-CA or pCAG-GFP.

### Electroporation of Tg(CRX-GFP) mice

A plasmid mixture, including 0.33 µg/µl pCAG-dGBP1x2-Flpo, 0.49 µg/µl pCAFNF-DsRed and 0.66 µg/µl pCAG-nlacZ were injected into Tg(CRX-GFP) and wildtype littermates, with the person doing the injections blind to the genotype of injected pup. Electroporated retinas were harvested at P14.

### Retinal histology

Isolated mouse retinas were fixed at room temperature for 30 min in 4% paraformaldehyde (PFA)/phosphate buffered saline (PBS) solution. Retinas were then transferred to 30% sucrose in PBS, and subsequently into a 1:1 mixture of 30% sucrose/PBS and Optimal Cutting Temperature compound (OCT) for sectioning. 20 µm retinal cryosections were cut on a Leica CM3050 cryostat (Leica Microsystems).

## Retinal immunohistochemistry

Retinal cryosections were incubated in blocking solution (3% normal goat serum, 1% BSA, 0.1% Triton-X, 0.02% SDS in PBS) for 1 hr and stained for primary antibody overnight at 4°C. Immunostained cryosections were washed three times in PBS and stained for secondary antibodies in blocking solution for 2 hr at room temperature. Slides were then washed in PBS and mounted for imaging in Fluoromount-G (0100–01; Southern Biotechnology Associates, Birmingham, AL). Retinal section images were acquired on a Zeiss LSM780 confocal microscope, on a 40x oil immersion objective.

## Analysis

Electroporated and immunostained retinas were quantified as 20 μm thick retinal cryosections imaged via confocal microscopy. Regions of dense electroporation were selected for quantification. Quantification approaches were described previously (*Tang et al., 2013*).

## rAAV production and injections

rAAV (2/1) virus preparations were made from pAAV-EF1α-dGBP1x2-Flpo-NW and pAAV-CAG-FLEX$^{FRT}$-ChR2-mCherry. All rAAVs were injected in the range of $10^{13}$–$10^{14}$ genome copies/ml, assayed by PCR of rAAV vectors at Boston Children's Hospital (Zhigang He lab). Primers targeted the ITR region of AAV vectors, and were: Forward - 5'-GACCTTTGGTCGCCCGGCCT-3', Reverse - 5'-GAGTTGGCCACTCCCTCTCTGC-3'. Note that we found this titering method gave about 10-100 fold higher numerical value in titer than other titering methods.

## Brain injections and electrophysiology

### Intracranial virus injection

For AAV infection of cerebella, Tg(GAD67-GFP) mice and GFP-negative littermates of either sex aged 3–4 weeks were anesthetized with ketamine/xylazine/acepromazine at 100, 2.5 and 3 mg per kg of body weight, respectively, and a continuous level of deep anesthesia was maintained with 5% isoflurane. A total volume of 200 nl of the following viral constructs: rAAV-2/1-EF1α-dGBP1x2-Flpo-NW, rAAV-2/1-FLEX$^{FRT}$-ChR2(H134R)-mCherry, rAAV-2/8-ZsGreen (*Tang et al., 2015*) (GFP-negative mice only) were injected into cerebellar cortex using a stereotactic device. For some experiments, 20 nl of green fluorescent beads (Lumafluor) were injected instead of rAAV-2/8-ZsGreen to monitor successful injection. 3 weeks later, brain tissue was fixed for immunohistochemistry, or prepared for electrophysiology.

### Cerebellar histology

Mice were transcardially perfused with 4% PFA in PBS (pH = 7.4) and the brains were post-fixed overnight at 4°C in the same solution. Parasagittal vermal slices of the cerebellum were cut at 50 μm thickness on a Leica VT1000S vibratome. Slices were then mounted on Superfrost slides (VWR, Radnor, PA) using Prolong Diamond mounting medium (Invitrogen). Images were acquired with an Olympus FV1000 or FV1200 confocal microscope.

### Slice preparation for electrophysiology

Mice were anaesthetized with ketamine/xylazine/acepromazine at 200, 5 and 6 mg per kg of body weight. Anaesthetized mice were intracardially perfused and processed to generate parasagittal cerebellar slices for electrophysiology as previous described (*Tang et al., 2015*).

### Electrophysiological recordings

Slices were superfused with ~32°C warm ACSF at a flow rate of ~3 ml/min in a recording chamber heated by an inline heater (Warner instruments, Hamden, CT). PCs were visualized using an Olympus BX51WI microscope equipped with differential interference contrast (DIC). GFP+ and ChR2-mCherry + were imaged using a custom two-photon laser-scanning microscope with 750 nm illumination. Visually guided recordings were performed with ~2 MΩ (PCs) borosilicate glass pipettes (Sutter Instrument). The internal solution for voltage-clamp recordings contained the following (in mM): 140 cesium methanesulfonate, 15 HEPES, 0.5 EGTA, 2 TEA-Cl, 2 MgATP, 0.3 NaGTP, 10 phosphocreatine-tris2, and 2 QX 314-Cl (pH adjusted to 7.2 with CsOH). Recordings were performed with a 700B

Axoclamp amplifier (Molecular Devices) and were controlled with custom software written in Matlab (ScanImage, available on GitHub. Generously provided by B. Sabatini, Harvard Medical School). ChR2-mCherry+ cells were excited using a 473 nm wavelength blue laser (OptoEngine, Midvale, UT) coupled through the excitation pathway of the microscope. Laser light was focused onto slices through a 40x water-immersion objective. Brief light pulses (0.5–1 ms) at an intensity of ~3–10 mW/mm$^2$ evoked ChR2-mediated photocurrents. Light-evoked synaptic currents were blocked by bath application of the GABA$_A$ receptor blocker SR 95,531 (Tocris, Bristol, United Kingdom). For cerebellar data, statistical significance was assessed with one-way ANOVA. Statistical significance was assumed when $P < 0.05$.

## Quantifications

To quantify the specificity of rAAV-delivered dGBP1x2-Flpo in the brain, ChR2-mCherry+ cells were compared between GFP+ cells in Tg(GAD67-GFP) brains injected with rAAV-EF1α-dGBP1x2-Flpo-NW and rAAV-CAG-FLEX$^{FRT}$-ChR2-mCherry, and ZsGreen+ cells in wildtype brains injected with rAAV-EF1α-ZsGreen, rAAV-EF1α-dGBP1x2-Flpo-NW and rAAV-CAG-FLEX$^{FRT}$-ChR2-mCherry. To further rule out the confounding effect of injecting rAAV-ZsGreen into only the wildtype brains, we counted the number of ChR2-mCherry+ cells in Tg(GAD67-GFP) and wildtype whole brain sections that both were co-injected with green fluorescent beads (along with rAAV-EF1α-dGBP1x2-Flpo-NW and rAAV-CAG-FLEX$^{FRT}$-ChR2-mCherry) (**Figure 5—figure supplement 2D**). Both approaches tested for the specificity of rAAV-delivered dGBP1x2-Flpo.

## CRISPR experiment

The human LoxP-LacZ cell line was obtained from Allele Biotech (San Diego, CA) (SKU: ABP-RP-CLA-CLOXE), and cultured as instructed in the product manual. Cas9 activity was assessed by detecting βgal-expressing cells in wells transfected with pX330-dCC-Cas9 and either pCAG-C-CA or pCAG-GAPDH-AU1 control construct (simply called AU1 in the main text). In addition, pCAG-mCherry is included as a transfection marker. For X-gal staining, cells were fixed on ice with 0.5% Glutaraldehyde for 5min. X-gal staining was performed as previously described. Cells were left at room temperature overnight for color development. Images were acquired by Keyence BZ9000 microscope. The number of mCherry$^+$ and X-gal$^+$ cells was quantified by Fiji software. The normalized Cas9 activity is calculated by dividing individual replicate values of specific conditions by the average number of X-gal+ cells induced by pX330-loxPgRNA alone.

### HIV-1 Sensor experiments

#### Cell culture

ACH-2 (**Folks et al., 1989**) or CEM cells were cultured with 10 nM Phorbol 12-myristate 13-acetate (PMA) in complete RPMI 1640 medium for three days prior to transfection (**Folks et al., 1989**; **Fujinaga et al., 1995**). $3 \times 10^6$ cells were seeded in 6 well plates. Cells were kept in the same medium after transfection for two days. A total of 2 μg DNA was transfected with X-tremeGene HP (Roche, Basel, Switzerland) into each well. TagBFP transfection mixtures consisted of 0.25 μg plasmids of either CAG-driven αCA-Nb-TagBFP, αCA-dNb$^{6mut}$-TagBFP, or dGBP1-TagBFP, along with 1.75 μg pCAG-DsRed. TagRFP transfection mixtures consisted of 0.25 μg plasmids of either CAG-driven αCA-dNb$^{6mut}$-TagRFP or C-CA along with 1.75 μg of pCAG-GFP. Cells were fixed in 4% PFA for 30 min at room temperature. Cells were washed twice with 2% heat-inactivated fetal calf serum in PBS, and finally re-suspended in PBS to be used for flow cytometry. For immunofluorescence with flow cytometry, cells were fixed and permeabilized using the CytoFix/CytoPerm kit (BD Biosciences, San Jose, CA). The antibody KC57-RD1 (6604667, Beckman Coulter), which recognizes the 24 kDa protein, also known as CA, of HIV-1 core antigen, was used to detect CA.

#### Analysis

Flow cytometry data were analyzed using FlowJo (FlowJo, LLC, Ashland, OR). Cells were gated to remove dead cells and doublets. DsRed+ gate was determined by comparison to PMS-stimulated but un-transfected ACH-2 and CEM samples. Stimulated ACH-2 and CEM cells transfected with only pCAG-DsRed and filler plasmids were used to determine TagBFP- or TagRFP-negative signals to be gated out of TagBFP+ or TagRFP+ population. Between 70–350 gated DsRed+ cells were analyzed

per condition per experiment. The "% gated TagBFP cells given DsRed+ cell" parameter was determined by dividing the number of cells that were dual-positive for TagBFP and DsRed by the total number of gated DsRed cells and the "% gated TagRFP cells/transfected cell" parameter was determined by dividing TagBFP and GFP dual-positive cells by the total number of GFP+ cells. Fold induction was determined by dividing the total number of TagBFP+/DsRed+ or TagRFP+/GFP+ cells counted in ACH-2 cells by that counted in CEM cells. Two-tailed Mann-Whitney test assuming unequal variance was used for testing of statistical significance. $P < 0.05$ is judged as statistically significant.

## General microscopy and image analysis

### General information

Images were analyzed and processed on Imaris (Bitplane, Zurich, Switzerland), ImageJ (*Schneider et al., 2012*) and/or Adobe Photoshop software. Whenever possible, image settings were adjusted for saturation. Whenever samples were to be compared within an experiment, image settings and processing were kept constant. Imaris, Image J and/or Photoshop software were used for image processing and analysis. Images from in vivo electroporation were smoothened on Imaris using the median filter as 3x3x1 pixel dimension or on Photoshop using the blue function at 1 pixel. Image level was adjusted in Photoshop.

## Acknowledgements

We thank S Zhao and C Wang of the Z He laboratory (Boston Children's Hospital) for rAAV production (core service supported by grant NEI 5P30EY012196-17). We thank M Springer, B Huang, M Lichterfeld, E Scully, A Tsibris, D Kuritzkes and members of the Cepko/Tabin/Dymecki lab for input on the manuscript, and the Neurobiology Imaging Facility (supported by NINDS P30 Core Center grant NS072030) for consultation and instrument availability. We thank WG Regehr for research support. We thank the Dana Farber Flow Cytometry facility for assistance with FACS. We thank S Arber and Addgene for plasmids. In addition to direct funding, SR was funded by the US National Institutes of Health (R01 NS32405 and R01 NS092707 to WG Regehr).

## Additional information

### Competing interests

JCYT, ED, SW and CLC: Submitted a patent application regarding destabilized nanobodies. International Application No. PCT/US2016/027749 Priority: US Prov. Appl. No. 62/148,595. The other authors declare that no competing interests exist.

### Funding

| Funder | Grant reference number | Author |
|---|---|---|
| Howard Hughes Medical Institute | | Stephanie Rudolph |
| National Institutes of Health | F32 NS087708 | Binggege Guo |
| China Scholarship Council | | Jonathan Z Li |
| National Institutes of Health | K08 AI100699 | Jonathan Z Li |
| National Institutes of Health | R21 AI114448 | Constance L Cepko |

The funders had no role in study design, data collection and interpretation, or the decision to submit the work for publication.

### Author contributions

JCYT, Conceived the dNb idea during a mutual discussion, Initiated and coordinated all aspects of the project, Designed experiments, Performed the dNb screen, Performed and analyzed western blots, Conducted bioinformatics, Performed and analyzed cell culture and retinal electroporation experiments, Generated AAV reagents, Performed and analyzed mutation transfer experiments,

Performed and analyzed HIV-1 sensor experiments, Co-wrote the manuscript; ED, Conceived the dNb idea during a mutual discussion, Designed experiments, Performed the dNb screen, Performed western blots, Contributed to the writing of the manuscript; BE, Designed experiments, Performed and analyzed HIV-1 sensor experiments, Contributed to the writing of the manuscript; SR, Designed experiments, Performed brain injections, image acquisition and histology, Performed and analyzed optogenetics and neuronal physiology experiments, Contributed to the writing of the manuscript; BG, Performed and analyzed mutation transfer experiments, Performed and analyzed CRISPR/Cas experiments, Contributed towards the writing of the manuscript; SW, Designed CRISPR/Cas experiments, Performed and analyzed CRISPR/Cas experiments, Contributed towards the writing of the manuscript; EGE, Performed brain injections, image acquisition and histology, Contributed towards the writing of the manuscript; JZL, Designed experiments, Supervised HIV-1 sensor experiments, Contributed towards the writing of the manuscript; CLC, Designed experiments, Supervised the entire project, Co-wrote the manuscript

### Author ORCIDs
Jonathan CY Tang, http://orcid.org/0000-0002-2376-6901
Constance L Cepko, http://orcid.org/0000-0002-9945-6387

### Ethics

Animal experimentation: The Institutional Animal Care and Use Committee at Harvard Medical School approved all animal experiments conducted under protocols 428-R98, 04537 and 1493.

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
