## [Decision Letter]

Thank you for submitting your article "Detection and manipulation of live antigen-expressing cells using conditionally stable nanobodies" for consideration by *eLife*. Your article has been favorably evaluated by a Senior Editor and three reviewers, including Loren Looger and Liqun Luo, who is a member of our Board of Reviewing Editors.

The reviewers have discussed the reviews with one another and the Reviewing Editor has drafted this decision to help you prepare a revised submission.

Summary:

Tang et al. described a clever method to access and genetically manipulate cells expressing a specific antigen that can be recognized by a nanobody. Using FACS sorting of 293T cells infected with an MMLV library with random mutations in a GFP-binding nanobody, the authors identified a variant with 6 mutations that is only stable in the presence of endogenous GFP. Three of these mutations, which have the most destabilizing effect when assayed individually, are conserved among most of the characterized nanobodies against different antigens. The authors showed that transferring these mutations to a number of these nanobodies could confer them antigen-dependent stabilization. The authors have shown that by using these mutagenized nanobodies to Flp recombinases and Cas9, they can confer antigen-expressing cells Flp or Cas9 activity with high signal-to-noise ratio, and could allow optogenetic manipulation of GFP-expressing cells via Flp-mediated channelrhodopsin expression. The authors further demonstrate that a nanobody against HIV capsid protein can be used to monitor reactivation of HIV-infected cells by flow cytometry.

Overall this is an impressive study that will be of great interest to a general audience. It represents a significant improvement of genetic access to cells expressing GFP, an approach pioneered by the same group in two previous studies (Tang et al., 2013, 2015), in that it is more convenient (requires a single transgene), and more versatile in that it can be used to target different effectors (the Flp-DOG is immediately useful for a wide variety of applications). The ability to transfer the antigen-dependent stabilization to other nanobodies will find more applications.

Most of the reviewers' critiques can be addressed with textual changes, and some additional controls and data analysis. We are copying the full review below instead of integrating them into a single document. We want to emphasize the third point of Reviewer #1 and the first paragraph of Reviewer #2, both requesting the authors to have more even-handed discussion.

*Reviewer #1:*

1) The text and main figures can be improved so readers do not need to refer to Methods and figure supplements for basic information. For instance, it will be useful in the main text to say that the mutations in dGBP1 were generated by random mutagenesis, and to include the number of screened variants from the library. Figure 1—figure supplement 2 contains impressive in vivo characterization of the specificity of dGBP1-TagBFP, and I recommend that at least some of these panels be moved to main figure (can add one). Likewise, Figure 6 seems too concise and readers need to see supplement 1 for essential details – why not move some of the panels from supplement 1 to the main figure? It will be nice to expand the y-axis of Figure 6 so readers have a more detailed view of conditions 2-4 (the first condition can utilize a broken y-axis).

2) Figure 2 contains the only data about the generalizability of these destabilizing mutations into most other nanobodies. It will be useful for the readers to see not only "fold induction by antigen" as quantified, but also background TagBFP fluorescence, as the latter would affect the utility when the effector is a sensitive protein such as a recombinase.

3) While the method is powerful, the authors may want to add more caveats in their Discussion about the limitations. For instance, the subcellular compartment of the antigen distribution may affect its utility-a plasma membrane associated antigen is not going to be very useful for targeting a recombinase. Their in vivo results are mostly derived from a single, transgenically expressed GFP variant, etc.

*Reviewer #2:*

Recommendation: This is a nice, clever piece of work. All of the experiments are very carefully conducted and controlled. With some revisions (see below), and definitely a more even-handed Discussion of the limitations and true "killer apps" of the technique, rather than those that are more of a cute way to accomplish something that can be done already a different way (and sometimes more robustly), this paper should be very nice.

1) In the second paragraph of the Introduction: Nbs can indeed have "high" affinity and specificity, but by and large both of these are lower than for larger-format antibodies. This bears mentioning.

2) Results: “Strikingly, many variants showed fusion TagBFP aggregates within the cell when YFP is absent, but became soluble in the cytoplasm when YFP is present (data not shown)”, I think the authors should show this data in a supplementary figure This sort of information is critical for people trying to develop similar reagents to get a feel for how "soft" the engineering problem is.

3) I think the Discussion should be toned down a bit. It's great that they mention all the possible applications for dNbs, but they should also mention downsides versus other approaches. Many of the demonstrated applications are "toy" proof of principle, with many other more straightforward ways to accomplish the same end. Targeting endogenous proteins in non-genetic organisms is very intriguing, though, with few other immediate options. But the Discussion should be more even-handed to say that even though the data are great and it clearly works, these are not really experiments that one would find oneself doing.

4) Figure 1—figure supplement 1: Is that a significant difference in unmodified GBP1 +/- YFP? Does YFP compete for transcription/translation? Does this complicate the analysis throughout?

5) Figure 2: There are some increases in fluorescence by addition of antigen to unmodified Nb. Almost every one. This bears commenting.

6) Figure 2: The color plots shown for dNb don't look consistent with the bar plots. For instance, 3K7U looks tiny in color plot, ~7x on bar plot; 4EIZ looks big on color plot, ~4x on bar plot.

7) Figure 3: There's a ~3-4x decrease in GBP1-Cre with GFP. Steric hindrance?

8) Figure 4: Are the authors sure that the Cas9-gRNA is cutting at both loxP sites? A simple PCR could verify this and would help demonstrate the resilience of the fused enzyme.

9) Figure 6: The fluorescence of reconstituted αCA-TagBFP + CA is quite low. This leads to very weak gating in the FACS experiments.

*Reviewer #3:*

This paper introduces a novel molecular strategy for expressing arbitrary proteins specifically in cells that express GFP, but not in cells where GFP is absent. On the basis of a molecular screen the authors identify mutations within the scaffold of an anti-GFP nanobody, that make this molecular unstable by itself, but stable when bound to GFP. They demonstrate that this construct works both in vitro and in vivo when fused with a number of different proteins. They also show that the mutations introduced to GBP1 can be introduced to other nanobodies, where they induce conditional instability, suggesting that this strategy might be widely applicable. Multimerizing the destabilized nanobodies increases the differential in stability between the bound and unbound states for fusions with recombinase enzymes such as Cre and Flpo. Furthermore, these conditionally expressed recombinases demonstrate conditional activity both in vitro and in vivo, and can be used to mediate the conditional expression of optogenetic proteins. Finally, this conditionally stable probe can be used to detect the presence of an antigen that is produced with HIV infection, and thus detect HIV +ve cells.

Overall this paper describes a method that has the potential for many applications and that has been carefully tested in many different contexts. It has the potential to greatly magnify the usefulness of existing mouse lines, thereby reducing the need to generate more lines. In general, the experiments carried out are convincing and appear to have been done with care. However, addressing a number of issues could make this paper stronger:

1) What happens to GFP when it is bound by dGBP1? For the experiments in Figure 1 and Figure 2, it would be very useful to know if the levels of GFP were reduced, and if so, by how much?

2) One of the most exciting potential applications is to use dNbs against endogenous proteins. Although the differential between expression levels in the presence and absence of endogenous antigens is considerably less than with GFP (6-7x or less), it is possible that these could be improved with additional mutations or by multimerization, etc. However, if these dNbs mediate the elimination of targets, their potential utility would be greatly diminished. Please include data on what happens to the levels of endogenous targets following expression of dNbs.

3) The differential for GBP1 shown in Figure 2 is only ~10, with an extremely large error bar. Please explain why this number is so low, when data in other locations (e.g. Figure 1—figure supplement 1) suggest that it should be many times higher.

4) In the paper it is suggested that under certain circumstances this method is better than the one that was previously published for mediating GFP-dependent expression. Please show some data to directly compare the two methods.

---

## [Author Response]

Most of the reviewers' critiques can be addressed with textual changes, and some additional controls and data analysis. We are copying the full review below instead of integrating them into a single document. We want to emphasize the third point of Reviewer #1 and the first paragraph of Reviewer #2, both requesting the authors to have more even-handed discussion.

Reviewer #1:

*1) The text and main figures can be improved so readers do not need to refer to Methods and figure supplements for basic information. For instance, it will be useful in the main text to say that the mutations in dGBP1 were generated by random mutagenesis, and to include the number of screened variants from the library.*

We made the following changes to address these comments:

A) “We generated a Moloney murine leukemia virus (MMLV) library encoding randomly mutagenized variants of GBP1 fused to the blue fluorescent protein, TagBFP.”

B). “One hundred GBP1 variants were then individually screened for enhanced TagBFP expression in the presence of yellow fluorescent protein (YFP), a GFP derivative known to also interact with GBP1”.

C) We moved the retina electroporation experiment from Figure 1—figure supplement 2 to Figure 1 itself to help the reader follow the data more easily.

D) We moved the FACS plots from Figure 6—figure supplement 1 to Figure 6 itself to make it easier for the reader to follow the data.

Figure 1—figure supplement 2 contains impressive in vivo characterization of the specificity of dGBP1-TagBFP, and I recommend that at least some of these panels be moved to main figure (can add one).

We have now merged panels A-C from Figure 1—figure supplement 2 with Figure 1.

Likewise, Figure 6 seems too concise and readers need to see supplement 1 for essential details – why not move some of the panels from supplement 1 to the main figure? It will be nice to expand the y-axis of Figure 6 so readers have a more detailed view of conditions 2-4 (the first condition can utilize a broken y-axis).

We have modified the Figure 6 graph such that it is broken on the y-axis and therefore emphasizes conditions 2-4 better. We have moved part of the Figure 6—figure supplement 1 panels into Figure 6.

2) Figure 2 contains the only data about the generalizability of these destabilizing mutations into most other nanobodies. It will be useful for the readers to see not only "fold induction by antigen" as quantified, but also background TagBFP fluorescence, as the latter would affect the utility when the effector is a sensitive protein such as a recombinase.

We have displayed the background TagBFP fluorescence as colored bars in the heat map. We feel that this display conveys the type of background associated with each nanobody. To further help the reader, we have now included a source data excel file ([Supplementary-material SD1-data]) showing the raw% Nb-TagBFP fluorescence values from Figure 2.

3) While the method is powerful, the authors may want to add more caveats in their Discussion about the limitations. For instance, the subcellular compartment of the antigen distribution may affect its utility-a plasma membrane associated antigen is not going to be very useful for targeting a recombinase.

We have revised the Discussion as follows:

“As a possible caveat, the range of biological activities that could be driven by an endogenous protein may be limited by the protein’s natural functions and/or sub-cellular localization. […] Although we cannot predict the frequency with which Nb fusions will result in a phenotype, we believe that the method is strong enough to encourage its continued development and application. “

In addition to the reviewer’s specific point, we have also added other issues to consider:

“Destabilized nanobodies were originally developed with the desire to simplify the delivery of GFP-dependent reagents as well as to improve their performance.[…] Nevertheless, these considerations suggest that one should establish an appropriate level of construct delivery, such as the amount of DNA plasmid or virus to deliver for optimal Flp-DOG performance.”

Their in vivo results are mostly derived from a single, transgenically expressed GFP variant, etc.

We feel that the work with GFP lines is quite extensive. We actually validated the system using two different transgenic GFP lines (Tg(CRX-GFP) and Tg(GAD67-GFP)) in two different nervous system tissues (retina and brain), using two different gene delivery methods (electroporation and rAAV infection). We further demonstrated the practical utility of Flp-DOG for a practical purpose ex. Optogenetic manipulation of specific cell types in the brain.

Reviewer #2:

Recommendation: This is a nice, clever piece of work. All of the experiments are very carefully conducted and controlled. With some revisions (see below), and definitely a more even-handed Discussion of the limitations and true "killer apps" of the technique, rather than those that are more of a cute way to accomplish something that can be done already a different way (and sometimes more robustly), this paper should be very nice.

*1) In the second paragraph of the Introduction: Nbs can indeed have "high" affinity and specificity, but by and large both of these are lower than for larger-format antibodies. This bears mentioning.*

We were not able to identify publications that demonstrated that Nbs have lower affinity and specificity than larger-format antibodies. In contrast, we note at least one study showing that Nbs have similar affinity and specificity as larger-format antibodies, such as Fab fragments. This study also was cited in a book on single-chain antibodies and fragments. Here is the relevant passage:

"Despite the single domain nature of Nanobodies with three antigen binding loops in the paratope of which the CDR3 is the most important, their antigen-binding surface is as large as that of a scFv where the paratope is equally spread over the CDRs of the VH and VL domains. As a result the antigen specificity and the affinity or kinetic binding properties of a VHH or a scFv with its cognate antigen are within the same range."

As we were unable to find a study that shows that Nbs have lower affinity and specificity than larger format antibodies, we are not including such a statement.

2) Results: “Strikingly, many variants showed fusion TagBFP aggregates within the cell when YFP is absent, but became soluble in the cytoplasm when YFP is present (data not shown)”, I think the authors should show this data in a supplementary figure. This sort of information is critical for people trying to develop similar reagents to get a feel for how "soft" the engineering problem is.

We have now included an example of a GBP1 variant that shows a particularly strong aggregation phenotype in the absence of YFP co-expression. This is now shown in Figure 1—figure supplement 1. To be more accurate, we modified our statement about GBP1 variant aggregation phenotypes. The new statement reads as follows:

“Some variants showed fusion TagBFP aggregates within well-transfected cells when YFP was absent, but became soluble in the cytoplasm when YFP was present (Figure 1—figure supplement 1)”.

To further elaborate, the aggregation phenotype is not as widespread amongst GBP1 variants as we had originally conveyed. Aggregation was only seen in cells that overexpressed certain GBP1 variant-TagBFP fusion constructs. Since we have shown that one can modify a nanobody to eliminate aggregation issues (dGBP1), and that aggregation only occasionally occurs with certain GBP1 variants, it is entirely feasible to engineer or screen for improved variants without the aggregation phenotype in future studies.

3) I think the Discussion should be toned down a bit. It's great that they mention all the possible applications for dNbs, but they should also mention downsides versus other approaches. Many of the demonstrated applications are "toy" proof of principle, with many other more straightforward ways to accomplish the same end. Targeting endogenous proteins in non-genetic organisms is very intriguing, though, with few other immediate options.

We made significant attempts to tone down the Discussion, to mention the existence of other methods to accomplish the same thing. As listed above in response to Reviewer #1, we have added a discussion of caveats. We include below additional points along these lines.

Here are the discussion points we added.

“Additional improvements may be made to enhance the response of dNb fusion constructs to antigen co-expression. […] Possible optimization steps may include exploring different fusion orientations, linker lengths or linker compositions”.

But the Discussion should be more even-handed to say that even though the data are great and it clearly works, these are not really experiments that one would find oneself doing.

We disagree with the assertion that the experiments are not ones that one would find oneself doing, and would ask that the Reviewer be more specific about which experiments one might not find oneself doing. In terms of even-handedness, in the initial submission, we discussed some of the caveats, e.g. of background activity, and we have now expanded upon this discussion, as described above. It would seem that the Reviewers recognize that many of the experiments presented establish the utility of the method in detecting GFP, endogenous, or pathogen-specific proteins and using these proteins to allow manipulations.

Regarding detection assays, we are not aware of any reagent that allows one to detect and isolate cells (without background issues) based on the presence of intracellular viral proteins. One has to resort to cell fixation and permeabilization followed by antibody staining of dead cells, or perhaps one can create a specific live cell reporter of infection, but we don’t know of a generalizable way to do this, given the need to identify virus-specific promoters, which are harder to identify than the method that we describe using viral protein sequences.

Using two dNbs to endow a fusion protein with coincidental detection specificity is not something that one would not find oneself doing. Intersectional strategies to target cell populations based upon their combinatorial molecular expression have been developed and are in use, with the most popular approach in the mouse community focused around intersection of Cre and Flp activity. Thus, this concept is not a frivolous one with no precedent.

The ability to integrate destabilized nanobodies with CRISPR will enable one to perform cell-targeted genome editing in cases where one does not have an appropriate cell type-specific promoter to specify Cas expression.Demonstration of such an approach will greatly interest many who wish to restrict genome editing to specific cellular conditions and/or cell types. In terms of even-handedness, we again will ask the Reviewer to be more specific as to where he or she finds that we have over-reached.

4) Figure 1—figure supplement 1: Is that a significant difference in unmodified GBP1 +/- YFP? Does YFP compete for transcription/translation? Does this complicate the analysis throughout?

We initially did not carefully measure protein levels of unmodified GBP1-TagBFP expression with or without YFP. The quantification scheme used in Figure 1—figure supplement 1 was not the best for identifying changes in protein level, since it involved counting the number of GBP-TagBFP-positive cells rather than measuring GBP-TagBFP protein level in cells. We now include a more carefully performed western blot experiment analyzing YFP’s effect on GBP1-TagBFP protein level (shown in newly introduced Figure 1—figure supplement 1; included in earlier response to Reviewer #2). We found a stabilizing effect from YFP co-expression.

As far as we can tell, there is no reason to suspect that YFP is competing for GBP1 transcription/translation.

5) Figure 2: There are some increases in fluorescence by addition of antigen to unmodified Nb. Almost every one. This bears commenting.

We have now noted the YFP-dependent stabilization of GBP1 effect in the text as well as in a new western blot experiment in Figure 1—figure supplement 1:

“Interestingly, we detected an enrichment of wildtype GBP1-TagBFP protein in the presence of YFP (Figure 1—figure supplement 1).“

While some replicates did show increases in fluorescence as a result of antigen co-expression, this trend mostly did not hold up across three independent experiments for each unmodified Nb. Thus, the slight increases in single experiments could not be used to conclusively infer stabilization of unmodified Nb by antigen. To further help the reader, we have now included a source data excel file ([Supplementary-material SD1-data]) showing the raw% Nb-TagBFP fluorescence values from Figure 2.

6) Figure 2: The color plots shown for dNb don't look consistent with the bar plots. For instance, 3K7U looks tiny in color plot, ~7x on bar plot; 4EIZ looks big on color plot, ~4x on bar plot.

This is due to the fact that the background fluorescence activity (without antigen co-expression) is different between the two nanobodies. 3K7U has less background fluorescence than 4EIZ. The high background activity for 4EIZ makes the fold induction less than others that have a similar level of fluorescence in response to antigen co-expression. To further help the reader, we have now included a source data excel file ([Supplementary-material SD1-data]) showing the raw% Nb-TagBFP fluorescence values from Figure 2.

7) Figure 3: There's a ~3-4x decrease in GBP1-Cre with GFP. Steric hindrance?

Some sort of steric hindrance may indeed be contributing to the slight decrease in GBP1-Cre activity with GFP co-expression. We have made a comment about this in the Discussion section (optimization of antigen-dependent sensors and effectors):

“Additional improvements may be made to enhance the efficiency of dNb fusion constructs in response to antigen expression. […] Possible optimization steps include exploring different fusion orientations, linker lengths or linker compositions.“

8) Figure 4: Are the authors sure that the Cas9-gRNA is cutting at both loxP sites? A simple PCR could verify this and would help demonstrate the resilience of the fused enzyme.

The Cas9-gRNA complex probably generates a combination of cuts at one or both loxP sites in the engineered genomic locus. The design of the assay is such that β-galactosidase activity would only appear upon excision of the loxP-STOP-loxP cassette from the engineered genome, and this would most likely occur after both loxP sites were cut. Although this was not shown in the form of PCR, the experimental controls sufficiently demonstrated the specificity of the system for Cas9 as well as gRNA expression. Without a gRNA, β-galactosidase activity was not detected, demonstrating a strong casual relationship between gRNA expression and STOP cassette removal.

*9) Figure 6: The fluorescence of reconstituted* α

CA-TagBFP + CA is quite low. This leads to very weak gating in the FACS experiments.

We acknowledged this in the Discussion and offered possible ways for optimization:

“Additional improvements may be made to enhance the response of dNb fusion constructs to antigen co-expression. […] Possible optimization steps may include exploring different fusion orientations, linker lengths or linker compositions.”

Reviewer #3:

Overall this paper describes a method that has the potential for many applications and that has been carefully tested in many different contexts. It has the potential to greatly magnify the usefulness of existing mouse lines, thereby reducing the need to generate more lines. In general, the experiments carried out are convincing and appear to have been done with care. However, addressing a number of issues could make this paper stronger:

*1) What happens to GFP when it is bound by dGBP1? For the experiments in Figure 1 and Figure 2, it would be very useful to know if the levels of GFP were reduced, and if so, by how much?*

We agree with the reviewer about the usefulness of such data. We performed an experiment to test whether dGBP1-TagBFP expression affected YFP protein level. Compared to a destabilized anti-CA-TagBFP negative control protein, we did not detect any clear difference in YFP level regardless of dNb-TagBFP identity. We have included this data in Figure 2—figure supplement 3. We note that this issue may be idiosyncratic for each fusion protein and its target antigen. In addition, it might depend upon one’s experimental conditions, e.g. when dNb is expressed in large excess relative to the antigen.

*2) One of the most exciting potential applications is to use dNbs against endogenous proteins. Although the differential between expression levels in the presence and absence of endogenous antigens is considerably less than with GFP (6-7x or less), it is possible that these could be improved with additional mutations or by multimerization, etc. However, if these dNbs mediate the elimination of targets, their potential utility would be greatly diminished. Please include data on what happens to the levels of endogenous targets following expression of dNbs.*

We did not perform the experiment suggested by the Reviewer as we do not believe that any one nanobody-fusion and its cognate antigen will give data that are predictive of the activity of other antigens, or of phenotypic effects, for the reasons cited below. We have also added the following paragraph into the Discussion:

“There could be cases where antigen-Nb interactions lead to less antigen activity. Reduction in activity will depend on several variables. […] Although we cannot predict the frequency with which Nb fusions will result in a phenotype, we believe that the method is strong enough to encourage its continued development and application.”

3) The differential for GBP1 shown in Figure 2 is only ~10, with an extremely large error bar. Please explain why this number is so low, when data in other locations (e.g. Figure 1—figure supplement 1) suggest that it should be many times higher.

We also noticed the large error bar. We have now repeated and re-analyzed 3 independent experiments in Figure 2 related to GBP1. The new results are much more consistent with visual expectations. Several factors likely contributed to the originally large error bar. The primary causes were transfection and/or imaging. The extremely low background activity for dGBP1-TagBFP in the absence of YFP co-expression made it hard to distinguish signal from background. The low background activity made it extremely sensitive to unevenness in auto-fluorescence in the fields that were imaged. Since we presented a ratio of fluorescence in the presence and absence of antigen, differences in background activity lead to large differences in fold induction. We should have paid more attention to this particular one, but we were happy with the level of dGBP1-TagBFP fluorescence induction in the presence of YFP, i.e. the fluorescence in the presence of antigen was always robust. We have now done a better job of measuring background consistently across the tissue culture dish, which improved the error.

4) In the paper it is suggested that under certain circumstances this method is better than the one that was previously published for mediating GFP-dependent expression. Please show some data to directly compare the two methods.

It is difficult to directly compare the activity of the dimerization-based systems with that of the destabilization system, since interpretations are confounded by the fact that different numbers of components have to be delivered to a cell for each method. In other words, one has to take account the differences in efficiency of delivering the full set of required constructs to single cells, as well as their ratio, in order to compare performance of the different systems. This very problem actually serves to illustrate the improvements achieved with the destabilization approach.

However, we can ask what is the efficiency of turning on a fluorescent reporter for each method. All components required for activation of the system were delivered by electroporation into the mouse retina under similar conditions. We found that the efficiency of GFP-dependent TagBFP reporter activation with dGBP1 was 90-100% (Figure 1—figure supplement 2), whereas for both T-DDOG and CRE-DOG systems, the efficiency of activating a TdTomato reporter was around 60% (see Figure S4B and S4E in Tang et al. 2013 and Supplementary Figure 2h in Tang et al. 2015). In the T-DDOG paper, the drop in efficiency is likely due to the need to deliver all four necessary plasmids to single cells. After this factor was taken into account, the T-DDOG method achieved at best around 93% efficiency. One might appreciate from this example, that a method that requires delivery of a single (or at most two) components will be easier to use, and will work more efficiently, than a method that requires delivery of a greater number of components.

We have included a statement of this in the Results section:

“Strikingly, the efficiency of TagBFP stabilization by GFP expression was nearly 100%, i.e. almost every GFP+ cell was TagBFP+ (Figure 1—figure supplement 2). The efficiency of the TDDOG and CRE-DOG systems was, at the highest, ~60% in similarly designed electroporation experiments ^1, 2^. This difference likely reflects the necessity of delivering a greater number of components for T-DDOG and CRE-DOG experiments.”

And we have included a more comprehensive discussion of the pros/cons of the GFP/dGBP1 system in the Discussion, as noted in the response to Reviewer #1 above.